# Understanding Insulin in the Age of Precision Medicine and Big Data: Under-Explored Nature of Genomics

**DOI:** 10.3390/biom13020257

**Published:** 2023-01-30

**Authors:** Taylor W. Cook, Amy M. Wilstermann, Jackson T. Mitchell, Nicholas E. Arnold, Surender Rajasekaran, Caleb P. Bupp, Jeremy W. Prokop

**Affiliations:** 1Department of Pediatrics and Human Development, College of Human Medicine, Michigan State University, Grand Rapids, MI 49503, USA; 2Department of Pharmacology and Toxicology, Michigan State University, East Lansing, MI 48824, USA; 3Department of Biology, Calvin University, Grand Rapids, MI 49546, USA; 4Office of Research, Corewell Health, Grand Rapids, MI 49503, USA; 5Division of Medical Genetics, Corewell Health, Grand Rapids, MI 49503, USA

**Keywords:** insulin, genomic variants, expression, splicing, protein folding, protein processing, receptor binding

## Abstract

Insulin is amongst the human genome’s most well-studied genes/proteins due to its connection to metabolic health. Within this article, we review literature and data to build a knowledge base of Insulin (*INS*) genetics that influence transcription, transcript processing, translation, hormone maturation, secretion, receptor binding, and metabolism while highlighting the future needs of insulin research. The *INS* gene region has 2076 unique variants from population genetics. Several variants are found near the transcriptional start site, enhancers, and following the *INS* transcripts that might influence the readthrough fusion transcript *INS–IGF2*. This *INS–IGF2* transcript splice site was confirmed within hundreds of pancreatic RNAseq samples, lacks drift based on human genome sequencing, and has possible elevated expression due to viral regulation within the liver. Moreover, a rare, poorly characterized African population-enriched variant of INS–IGF2 results in a loss of the stop codon. *INS* transcript UTR variants rs689 and rs3842753, associated with type 1 diabetes, are found in many pancreatic RNAseq datasets with an elevation of the 3′UTR alternatively spliced *INS* transcript. Finally, by combining literature, evolutionary profiling, and structural biology, we map rare missense variants that influence preproinsulin translation, proinsulin processing, dimer/hexamer secretory storage, receptor activation, and C-peptide detection for quasi-insulin blood measurements.

## 1. Introduction

Diabetes or prediabetes impacts 1 in 3 individuals within the United States [1], costing $237 billion each year, or 1 of every 4 dollars spent on health care within the United States [2]. Diabetes mellitus is diagnosed as hyperglycemia higher than the threshold blood glucose concentration, resulting in microvascular end-organ damage. Hyperglycemia is the sequelae of numerous pathophysiological processes resulting from either the pancreatic beta-cells’ inability to secrete enough insulin or their systemic resistance to insulin.

Type 1 diabetes (T1D) is associated with the decreased production of insulin through the destruction of pancreatic beta cells [3,4], requiring expensive, lifelong recombinant insulin injections [5]. Genome-wide association studies (GWAS) and epigenetic profiling have yielded insights into T1D risk factors of the immune system, such as the contributions of T-cells, pancreatic acinar, and ductal cells to disease pathology [6]. In multiple independent studies, the rs689 variant of the insulin gene (*INS*) was identified as the most significant locus for T1D risk when masking the large effect size of HLA loci [6,7,8].

Rare genomic variants within *INS* and other genes have recently been identified to elevate odds for T1D and neonatal/monogenic diabetes [8,9]. Monogenic diabetes by itself is uncommon. Advances in various molecular techniques and our immune understanding of T1D may one day lead to a cure for T1D through increasing the body’s natural insulin production [10]. The drive to develop precision therapy wherein the individual patient is better managed requires a comprehensive understanding of diverse axes of biology. These axes include homeostatic context, genomic variation, chromatin signals that mark genes as active or repressed in tissues, expressed transcripts, and increased knowledge of lifestyle/environmental risk factors. However, diagnoses, treatment options, and investments in T1D insights come with an increasing socioeconomic and racial disparity that must be addressed [11].

Contrary to T1D, Type 2 diabetes (T2D) is associated with a developed resistance to insulin, accounting for ~90% of individuals with diabetes [12,13,14]. T2D results in the elevation of blood glucose, which, if uncontrolled, can result in pathologies including myocardial infarction, stroke, amputation, renal disease, eye damage, neuropathy, and increased tumor burden [15]. Public health strategies have struggled to slow the epidemic, even in countries with the greatest financial and scientific resources. There are 12 drug classes currently approved by the U.S. Food and Drug Administration (FDA) to control blood glucose and modify disease course, but these drugs do not provide a cure or result in the remission of disease. These agents are often prescribed based on nonmedical considerations such as cost, patient preference, or comorbidities not accounting for biological mechanisms. Thus, balancing glucose levels through diet or pharmacology (Metformin, sulfonylureas, meglitinides, pioglitazone, and α-glucosidase inhibitors) within T2D patients is of top priority [16]. However, T2D is undiagnosed or uncontrolled in many individuals, with socioeconomic and racial disparity playing a role [17,18,19]. In advanced or uncontrolled T2D, insulin production is often hindered through decreased pancreatic beta cell mass, yielding similarities to T1D [20]. Genetic variants in genes involved in insulin gene regulation, protein processing, and secretion were identified as risk factors for T2D [21,22].

In addition to the complex system of genetic variance that can affect diabetes, an individual’s environment can also play a significant role. For example, an increase in body mass index and diagnosis of obesity can disrupt the endocrine pathways that increase diabetes incidence [23]. Geographical areas with limited access to nutritious food options, called food deserts, were linked to increased obesity and diabetes, including among youth [24,25]. Low-income neighborhoods often lack outdoor areas and facilities for residents to safely engage in physical activity, exacerbating health issues and making it difficult for even motivated residents to exercise [26]. Air pollution from highways contributes to poor cardiovascular health and may also directly contribute to an increased risk for obesity and diabetes [26,27,28].

In attempts to improve community health, some urban planners have begun to advocate for a focus on creating ‘walkable neighborhoods’, designing new city blocks, and even removing old infrastructure in a way that encourages pedestrian and bicycle travel and ensures easy access to parks and quality grocery stores [27]. An analysis of existing neighborhoods shows that more walkable neighborhoods are associated with lower obesity rates, and that residents also had lower predicted ten-year cardiovascular disease risks than their counterparts in less walkable neighborhoods [28,29]. Many population groups with a high risk of diabetes live in communities disproportionately impacted by urban planning, air pollution, low income and food deserts. We must carefully interpret the genetic component of diabetes risk based on the inheritance of functional variants relative to cultural inheritance and community dynamics [30]. Here, we provide a genomic dissection of *INS* variants as an example of functional interactions between genetics and environment, which might apply to other metabolic genes.

The decreasing cost of whole genome sequencing and the broader application of these techniques [31] has provided new opportunities to investigate the genetic factors associated with T1D and T2D. Larger-scale sequencing projects such as gnomAD [32], All of Us [33,34], and the UK biobank [35] have yielded diverse racial/cultural genomes that can be connected to medical insights for diabetes diagnosis, treatments, and complications. Genetic studies can inform the risk of diabetic complications but often rely on GWAS and less on an individual’s genome, yielding many cultural/population disparities [31]. Multiple forms of neonatal/monogenic diabetes were identified using advanced genome sequencing strategies [36,37,38], which can be applied independent of socioeconomic and population background through which GWAS can be biased. In addition, a more complex polygenetic understanding of diabetes can be devised for each individual through a network of genes/variants involved in beta cell function and insulin mechanisms of action, which can be integrated with medical data to determine a patient’s diabetes risk score [39,40,41].

The advancement of sequencing has discovered thousands of variants connected to insulin, many with little characterization. Characterizing the nature of variants within *INS* and those that affect proteins involved in the synthesis and secretion of insulin is vital to achieving personalized diabetic treatment. This article summarizes all known variants, both rare and common, of the *INS* gene while highlighting genes/proteins that might be associated with the altered synthetic capacity of insulin. The article reflects a literature and data review of *INS*. This ranges from how variants influence *INS* expression and splicing to how missense variants contribute to insulin misfolding, mislocalization, and receptor binding alterations.

## 2. Results

### 2.1. INS Genomic Location

The *Insulin* gene (*INS*) was found on chromosome 11 in the p15.5 region (Figure 1A). It is found between *Tyrosine Hydroxylase* (*TH*), *Insulin-like Growth Factor 2* (*IGF2*), and *IGF2 noncoding antisense transcript* (*IGF2-AS*) [42], as can be seen in Figure 1B. Several known splicing derivatives exist for each gene, with *IGF2* having multiple known transcriptional start sites (TSS), and there is a known merged transcript of *INS* and *IGF2* (*INS–IGF2*) resulting from transcriptional readthrough. The protein coded by *INS–IGF2* is a 200 amino acid protein (UniProt F8WCM5) composed of the first 62 amino acids of insulin followed by a frameshift that differs from the IGF2 coding sequence. This transcript was associated with insulinomas and autoimmunity [43,44]. The sequence of *INS* and *INS–IGF2* exons was well-conserved throughout vertebrates (Figure 1B, conservation). While there are many known datasets for gene regulation in this region and thousands of human genomic variants, little work has been done so far to integrate this knowledge into the complex regulation of beta cells and *INS* expression. One of the strongest gene regulation datasets for pancreatic islets comes from the Roadmap Epigenomics project [45], where the measurement of six different epigenetic marks from 98 different human cells/tissues (including pancreatic islets) annotates transcriptional regulation and multiple enhancers near *INS* in islet cells (Figure 1B, 18-state Genome Annotation). Transcription factors, including PDX1, NEUROD1, and MAFA, are known to control pancreatic beta cell development and *INS* transcription [46,47,48,49].

Therefore, we further zoomed into the *INS* islet regulation region (hg19-chr11:2,176,237–2,182,988) to extract functional genetic variants (Figure 1C). Several GWAS loci were found in this region for T1D (rs3842753, rs689), T2D (rs3842770, rs571342427), severe autoimmune type 2 diabetes (rs3842753), glucose levels (rs3842753, rs3842752), and IGF1 levels (rs3842752) based on the GWAS catalog (Figure 1C, red text) [8,50,51,52,53,54,55,56,57,58,59,60]. It is somewhat surprising, however, that no variants were connected to *INS* expression levels within this region through the expression of quantitative trait loci (eQTLs). The eQTL strategy links common variants to the dysregulation of transcription within the GTEx consortium [61]. Variants in this region were only associated with altered *TH* expression (Figure 1C, eQTLs), which is likely due to the underpowered pancreas dataset of GTEx that prevents *INS* eQTL mapping.

We extracted genomic variants for the entire region of Figure 1C (hg19 chr11: chr11:2,176,237–2,182,988) from multiple sources, including the dbSNP database of common alleles [62], gnomAD genomes database of diverse genomes [32], Avada literature-extracted variants [63], ClinVar database of clinical rare disease genomes [64], and UniProt protein annotated variants [65]. The gnomAD allele frequency for different population backgrounds of these variants is available at https://doi.org/10.6084/m9.figshare.21917124.v1 (accessed on 6 December 2022). All variants were processed for transcription factor binding site (TFBS) alterations, transcript alterations, and protein changes.

### 2.2. Variants near INS with Genome-Wide Association Traits

The rs689 variant (https://genetics.opentargets.org/variant/11_2160994_A_T (accessed on 6 December 2022) is the most significant variant from GWAS for T1D [6,7,8,66,67] and is associated with latent autoimmune diabetes in adults [68,69,70] and atypical T2D [71,72]. This variant is highly associated with the development of islet autoantibodies to insulin, likely contributing to the initial development of T1D [73,74,75,76]. The variant is associated with B-cell anergy alteration [77]. However, it is not significantly associated with alterations in regulator T-cells [78].

rs689 was found in linkage disequilibrium with most of the T1D GWAS catalog variants found near *INS* (Figure 2A), including the highly correlated rs3842753 (R^2^ of 1) and rs3842727 (R^2^ of 0.89). The *p*-values of rs3842727 for various traits were far below that of rs689, suggesting that rs3842727 is not the lead SNP for functionality. However, the *p*-values of rs3842753 and rs689 for various traits were highly similar, with subtly more significance in rs689 for traits (Figure 2B,C). Outside of T1D, rs689/rs3842753 is associated with insulin-like growth factor 1 levels (IGF-1) [79,80], paternally inherited IGF-2 levels at birth [81], age of diagnosis for T1D [82], starting insulin within one year of diagnosis [83], size at birth [84], hemoglobin A1c levels [79], blood glucose levels [79], and the age at which diabetes was diagnosed [83].

These two variants are found throughout all populations but show enrichment within Japanese, Asian, Vietnamese, Korean, and South Asian populations, with the lowest levels observed in African/African American populations (Figure 2D). Both variants fall at potentially functional locations. rs689 was found within the 5′UnTranslated Region (5′UTR) of one *INS* transcript and near the 5′UTR splice site of the rest of the transcripts (Figure 2E). In silico and biochemical assays have suggested that the change does result in ex vivo splicing changes by the U2 small nuclear ribonucleoprotein [85,86,87], yet to the best of our knowledge, this has not been observed within human samples. rs3842753 was found within the 3′UTR for most of the *INS* transcripts (Figure 2E). We performed additional data analysis with various tools to further add annotation to this and all other *INS* variants.

### 2.3. Variants near INS from Genomic Databases

Around the *INS* gene are 2076 unique gnomAD variants from 76,156 genome sequences of humans with diverse population backgrounds [32]. Each of these variants was run through the Variant Effect Predictor (VEP) [88] for annotations of transcripts and regulation (Figure 3A,B). The allele count of variants is highly variable, with 30 unique variants found with >1000 allele counts in gnomAD and 6134 unique annotations from VEP (Figure 3A). Each variant can have multiple annotations due to the multiple transcripts of *INS* and *INS–IGF2.* Of VEP annotations, 61.4% were downstream of a transcript, 15.9% were upstream of a transcript, 10.5% were within introns, 4.7% were within noncoding transcripts, 3.5% were within regulatory regions, 1.1% were missense, 1.1% were 5′UTR, 0.6% were synonymous within a transcript, and 0.5% were 3′UTR (Figure 3B). In addition to gnomAD, we also addressed the outcomes of variants from ClinVar (pathogenic variants are in red, likely pathogenic variants are in orange, and variants of uncertain significance (VUS) are in gray), UniProt (cyan), and Avada (magenta), with the majority resulting in missense changes (Figure 3C).

To annotate noncoding variants, we utilized various datasets of gene regulation, including conserved TFBS [89], ENCODE transcription factor binding sites [90], JASPAR transcription factor binding profiles [91], and RegulomeDB [92]. These metrics were plotted relative to genome location and gnomAD allele counts (Figure 3D). The RegulomeDB variants of 2a and 2b fall within potential TFBS that overlap foot-printing assays of gene regulation and within gene regulation sites of the Roadmap Epigenomics initiative [45], and they are likely to impact transcriptional regulation. Those within conserved TFBS likely impact critical binding events for gene regulation. Addressing these two groups, several variants were found near the TSS of *INS* (Figure 3E) or within two downstream regulation sites (Figure 3F–G).

The *INS* TSS region contained both common and rare variants from gnomAD (Figure 3E). The most common variant within the TSS, rs3842738, had a gnomAD allele count of 27,653 and was found in all populations. The variant has a weak significant association with ubiquitin carboxyl-terminal hydrolase 8 levels [93]. The variant rs3842738 had a RegulomeDB score of 2b, falling within a GTF2I binding site and active TSS within endocrine pancreas tissue. Other RegulomeDB-found potentially impactive variants include rs915076855 (nine allele count, SP1 binding site) and rs1262751704 (two allele count, GATA2 in conserved TFBS), both of which are rare. Three rare variants of gnomAD fell within conserved TFBS, including rs1157485249, rs557938186, and rs1456326746 within the TATA box region. The TATA box is a critical activation component bound by the TATA box binding protein (TBP) near a TSS that regulates the recruitment of polymerases for active transcription [94,95]. The conservation of this TATA box and rare variants raises questions about whether variants such as these could result in T1D or T2D due to heterozygous decreased/increased expression of *INS*. Accurately understanding gene regulation and rare variants’ influences on *INS* or other genes is one of the most critical future research needs.

Other rare variants of possible regulation can be found outside of the TSS. For example, a downstream regulation region contained conserved TFBS for HNF3B and forkhead transcription factors (Figure 3F), which are critical for islet and pancreas development [96,97]. Within this region is the African/African American-enriched variant rs116079895. Another downstream regulation region contains a conserved P300 TFBS and a variant rs11602347, which has a RegulomeDB score of 2b (Figure 3G). The variant overlaps ChiP-Seq peaks for both TAF15 and EGR1, falling within the footprint motif of EGR1. While EGR1 is well known to modulate insulin effects on various metabolic/endocrine cells [98], EGR1 can modify *INS* expression levels [99]. Various environmental factors, such as the hepatitis C virus, were shown to activate IGF2 by modulating *EGR1* regulation [100]. As the rs11602347 variant falls between *INS* and *IGF2*, this is an interesting potential EGR1 regulation site requiring further wet lab analysis. The variant is found in all population backgrounds and is significantly associated with serum 25-hydroxyvitamin D levels [101] and T1D [6,7,8], despite having only 0.27 R^2^ with rs689.

Next, we addressed the role of gnomAD variants that fell within *INS* or *INS–IGF2* transcripts using the VEP-generated CADD scores (Figure 3H), which integrate multiple datasets to calculate a variant’s deleteriousness [102]. The ENST00000381330 transcript was the primary *INS* transcript and had 48 variants annotated to it, including missense, splicing, 3′UTR, 5′UTR, and synonymous variants (Figure 3I). An additional 226 variants were annotated to other *INS* or *INS–IGF2* transcripts. The rs689 variant was found at a splice site of ENST00000381330 and within a 5′UTR of ENSG00000254647. rs3842741 was also found in the 5′UTR of ENSG00000254647. The high R^2^ SNP to rs689, rs3842753, and the closely located rs3842752 were found in the 3′UTR of ENST00000381330. The variant rs5505 was found in the 5′UTR of ENST00000381330 and is enriched within non-Finnish Europeans but has no known trait associations. Three rare missense variants were observed within ENST00000381330, including rs144093133 (p.G75D, CADD 15.63, 28 allele counts in African/African Americans), rs139264769 (p.S76N, CADD 3.89, 11 allele counts in African/African Americans), and rs535989053 (p.A2T, CADD 21.2, 2 allele counts in East Asians). We next turned to the analysis of RNAseq reads for *INS* expression levels and their correlation to common variants.

### 2.4. INS Transcripts and Splicing

We started by analyzing 17,382 human samples of 54 tissues of GTEx [61] for *INS*, *INS–IGF2*, and *INSR* (the Insulin Receptor). GTEx contains tissue RNAseq collected from relatively healthy individuals who often had lethal-based trauma and donated their samples to science. *INS* transcripts were relatively unique to the pancreas, whereas *INS–IGF2* was expressed in the pancreas and the liver (Figure 4A). *INSR* was expressed in all tissues with elevation in spleen and ovary samples.

There were five *INS*, two *INS–IGF2*, and seven *INSR* transcripts. ENST00000381330 (*INS-202*) was the highest expressed *INS* transcript and coded for a 110 amino acid protein. ENST00000397262 (*INS-203*) and ENST00000250971 (*INS-201*) also coded for the 110 amino acid protein but had 5′UTR alterations. Transcript ENST00000512523 (*INS-204*) had an early termination that resulted in a 98 amino acid protein. ENST00000421783 (*INS-205*) utilized alternative splicing within the ORF of *INS*, resulting in a frameshift of protein sequence, and is predicted to result in non-stop decay, a process where transcripts are degraded because they lack a stop codon [103]. *INS–IGF2* had two known transcripts: ENST00000397270 (*INS–IGF2-202*, 828 nucleotides), which coded for a 200 amino acid protein and ENST00000356578 (*INS–IGF2-201*, 1706 nucleotides), which also coded for the 200 amino acid protein but with multiple spliced 3′UTR exons, which is predicted to be degraded by nonsense-mediated decay (NMD).

Each of the five *INS* and two *INS–IGF2* transcripts were processed for GTEx pancreas expression (328 samples), breaking down the distribution of expression between male and female patients of various age groups (Figure 4B). The *INS* ENST00000381330 transcript showed the highest levels, balanced over all age groups and between sexes. It should be noted that the number of outliers increased with the age of the patients from whom the samples originated. The four other *INS* transcripts had far lower expression while also showing the presence of a few samples with outlier levels. The *INS–IGF2* ENST00000397270 was identified within the pancreas, whereas the NMD-regulated longer ENST00000356578 showed expression in the liver (Figure 4B).

To expand these insights further, we explored the expression of *INS* and *INS–IGF2* in RNAseq datasets of the pancreas and liver deposited into the NCBI SRA database. There were 22 BioProjects containing 816 samples with *INS* expression (Table 1). Of the *INS* transcripts, ENST00000381330 was broadly expressed, with a few samples having high levels of the other spliced forms of *INS* (Figure 4C). Those samples with below 90% use of ENST00000381330 for *INS* and still having a high level of *INS* normalized to the *INSR* (Figure 4D) included high uses of ENST00000250971 within sample SRR1299332 (young patient beta cells), SRR6048793 (male, 23-year-old, non-diabetic islets), and SRR1299342 (young patient beta cells). ENST00000421783, the noted transcript for degradation by nonstop decay, was used within sample SRR8081597 (female, 12-year-old, non-diabetic islets). As rare variants associated with neonatal diabetes were identified to induce additional *de novo* splice alterations [104,105], there is a further need for surveillance within neonatal diabetes for splice-altering *INS* variants.

Individual samples showed a high correlation (R^2^ = 0.7344, Figure 4E) of the *INS* and *INS–IGF2* transcripts, whereas several samples had a higher-than-expected expression of *INS–IGF2*. Most of the samples with high *INS–IGF2* were from BioProject PRJNA691365, which is an RNAseq of isolated human pancreatic beta-cells. Within the liver samples, *INS–IGF2* was elevated within a handful of samples, with noted higher levels of the NMD-regulated transcript (Figure 4F). It is interesting to note that of the top 20 samples with this elevated *INS–IGF2* NMD transcript, nine had a hepatitis B/C viral infection (SRR5237638, SRR8382611, SRR8382605, SRR5237640, SRR8382623, SRR5237643, SRR8382602, SRR8382621, SRR8382601).

NMD is a mechanism that degrades transcripts with splice sites within the 3′UTR, preventing the production of proteins that might be damaging to the cell due to nonsense mutations [106,107,108]. The NMD molecular components are regulated by various cellular pathways, such as endoplasmic reticulum (ER) stress and immune activation [109,110]. Within the pancreatic beta-cells, NMD regulation is influenced by inflammatory cytokines and ER stress, resulting in changes to insulin biosynthesis [111].

The NMD process is involved in protecting cells from nonsense mutations and disease states and was shown to be critical for regulating intracellular viruses [112]. Many viruses thus evolve mechanisms to suppress the NMD pathway [113]. Within T1D, growing evidence of viral infections, including enteroviruses, rubella, mumps, rotavirus, parvovirus cytomegalovirus, hepatitis, and the common cold, was associated with the development of beta-cell autoimmunity [114,115,116,117]. Animal models of infection confirmed this association [118]. The engraftment of virally infected beta-cells from one mouse to another confers the risk of T1D [119].

In tissues such as the liver, environmental factors can alter the NMD pathway, resulting in increased transcripts normally degraded by NMD. Their elevation increases the potential for non-alcoholic fatty liver disease, hepatocellular carcinoma, alcoholic liver disease, insulin resistance, and T2D [120,121,122]. The hepatitis C virus (HCV) inhibits the NMD process [123]. What is striking is the relationship between liver disease, diabetes, and HCV, where insulin resistance and signaling are known to be altered [124,125,126]. Nearly one out of three individuals with liver cirrhosis also have diabetes, with the majority suffering from hepatogenous diabetes (HD) [127]. HCV is one of the main factors for developing liver cirrhosis and HD [128,129,130]. HD is distinct from T2D, with normal fasting glucose and beta-cell dysfunction, and it is usually independent of body mass index [131,132,133]. Interestingly, many of the factors of immune alterations that are altered by viral NMD suppression, such as tumor necrosis factor-alpha, are also associated with HCV-induced HD [134]. The observation of such strong associations between the viral regulation of NMD, the viral mechanisms of both T1D and HD, the role of *INS–IGF2* in insulin autoantibody production, and the odd *INS–IGF2* NMD-regulated transcript expression warrants future investigations.

### 2.5. Variants within Insulin Transcripts

We developed a list of transcripts with each common variant identified to fall within *INS*. We identified using the main *INS* isoform, ENST00000381330, that rs3842753 was the predominant genotype of the RNAseq samples (58%, Figure 5A), followed by rs3842753/rs3842752 (25%), and rs3842753/Wild Type (WT, 7%). In our analysis, we did include a sequence with rs3842753 and rs3842752 in combination, as these are within 14 bases of each other and could be observed together in RNA sequence reads. However, we never saw any RNAseq reads where the two variants were physically connected, suggesting they segregate independently. Bringing each genotype group together with total *INS*/*INSR* normalized expression levels showed that samples with the rs3842753 variant were more likely to have outliers of high expression (Figure 5B). Interestingly, of the samples that were compound heterozygous for rs3842753 and rs3842752, there tended to be a higher frequency of rs3842752 within their ENST00000381330 transcripts. However, samples with the highest *INS*/*INSR* expressions tended to be closer to 50% variant frequency (Figure 5C).

The rs689 variant in LD with rs3842753 (Figure 2A) was present only in the ENST00000397262 transcript. Surprisingly, 85% of RNAseq samples had the combination of both variants (rs689/rs3842753) in this transcript (Figure 5D), suggesting the isoform is primarily coded only when variants are present. Those samples where the expression of ENST00000397262 was high relative to ENST00000381330 had the presence of both rs689/rs3842753 (red and orange, Figure 5E,F). This suggests that due to the high penetrance of signal in RNAseq, the alteration of the 5′UTR splicing change of ENST00000397262 was likely the result of rs689.

### 2.6. Insulin Signal Peptide Missense Variants

Synthesis of the insulin protein starts with the initiation of translation of an mRNA encoding preproinsulin, with many factors impacting this process and increasing the risk of disease [135]. *INS* transcripts engage with the ribosome to make a 110 amino acid protein. This protein is composed of the signal peptide, B-chain, C-chain, and A-chain in which all of these regions have conserved amino acids and known missense variants (Figure 6A). To aid in interpreting these variants, we assessed the PDB for structures of insulin, where there are a few structures of proinsulin and many structures of mature insulin consisting of the A- and B-chains (Figure 6B).

The first 24 amino acids coded for a signal peptide that is critical for incorporating insulin into the rough ER for the secretory pathway of exocytosis [136]. There are no structures of the signal peptide for full preproinsulin within a lipid membrane. Thus, the interpretation of clinical variants relies on knowledge of signal peptide biology. The signal peptide was formed by multiple conserved hydrophobic amino acids (MALWMRLLPLLALLALWGPDPAAA, Figure 6A). Once protein synthesis is started on a ribosome, the signal peptide can be inserted within a lipid bilayer of the rough ER [137]. These hydrophobic amino acids, when altered, can result in neonatal/monogenic or early-onset diabetes. These include ClinVar-identified pathogenic and likely pathogenic variants at R6, P9, and A24, with additional literature annotated leucine amino acid 13 changed to arginine (L13R) [138].

The insertion of preproinsulin into the lipid membrane of the ER can be accomplished in two ways: cotranslational translocation, in which the signal peptide synthesis by the ribosome drives membrane insertion, or posttranslational translocation, where translocons guide the fully synthesized signal peptide into the membrane [139]. Most signal peptides rely on the signal recognition particle (SRP) complex, which binds the signal peptide and couples the complex to a translocon [140]. The insulin signal peptide alone appears to have weak gating potential. It requires translocon assistance driven by the translocon-associated protein (TRAP) complex, which requires multiple prolines of the insulin signal peptide [141]. The deletion of TRAP components from organisms results in the disruption of insulin biosynthesis [142]. HSP70 and several other cochaperones interact to prevent the aggregation of cytosolic peptides waiting for translocation into the ER [143]. Once engaged with the SRP or TRAP complex, the peptide must be inserted into the ER via post-translation translocation [139]. The conserved polar basic arginine (R6) of preproinsulin is required to engage with the ER translocon SEC61 that assists in the signal peptide insertion within the lipid membrane [139]. Arginine mutations to cysteine or histidine at amino acid six are associated with neonatal or early-onset diabetes [144,145].

With many genes involved in ribosome engagement, the association of the signal peptide to the rough ER, translocation of the signal peptide, continued ribosomal synthesis, termination of ribosomal synthesis, and folding within the ER, it is logical that variants within many of these genes of the pancreatic beta-cells could be associated with diabetes. Not only can genetic variants impact this process, but the health of the ER can also be associated with dysfunction of insulin synthesis. More research is needed to understand preproinsulin of the ER in healthy versus diseased beta cells and how additional variants may impact this process [146].

### 2.7. Insulin Structural Missense Variants

After incorporating the signal peptide into the lipid membrane, the preproinsulin is fully synthesized by the ribosome. This ER membrane-bound protein can then fold and be processed. While a large portion of newly synthesized proteins are degraded within cells, insulin within the beta-cells is very efficiently produced with prevented degradation [147], suggesting the existence of some critical ER processes for protein protection with unknown roles of genetic variants on this process. Within the protein sequence of insulin, there is evidence of conservation regarding foldability, successful post-translational modifications, the formation of dimer/hexamer storage units, and successful activation of the insulin receptor (Figure 6, Figure 7 and Figure 8) [139,148].

Three disulfide bonds maintain insulin’s structure [149] where variants of 3/6 cysteines are found pathogenic or likely pathogenic within ClinVar for neonatal diabetes (C43G-VCV000021114.5, C96S-VCV000068730.1, C96Y-VCV000013387.6, C96R-VCV000918067.1, C109F-VCV001526011.1). All four cysteine amino acids involved in inter-chain disulfide bonds are known to be involved in monogenic diabetes based on the literature [145]. The formation of these bonds and, thus, the folding of insulin requires specific environments within the pancreatic beta-cells and their organelles [150]. Significant post-translational processing is needed with organelle-specific steps and stages of insulin trafficking [136]. The successful folding of ER-bound preproinsulin requires oxidoreductases and chaperone proteins to prevent disulfide-linked aggregate formation. The cytosolic processing of insulin due to truncation of the signal peptide results in a lack of proper disulfide bond formation and folding [151].

The ER-folded protein must then be trafficked through the Golgi apparatus and processed to mature insulin. The preproinsulin is found in the lumen of the rough ER, where signal peptidases cleave the signal peptide at amino acid 25 [152]. The timing of folding and cleavage relative to relocalization from the ER to the Golgi is rather complex, where variants that impact folding can result in retention within the ER [153]. Variants at amino acid 23 of the signal peptide, such as A23S, do not disrupt cleavage. The more conserved amino acid 24 with a changed S24D inhibits cleavage and results in ER retention [154]. Changes to amino acid 24 were associated with autosomal-dominant neonatal/monogenic diabetes [9,154,155,156]. A24D is one of three variants that accounts for 46% of cases of neonatal diabetes [145], with the advanced progression of secondary complications, including brain lesions [157].

Proinsulin consists of 3 chains: the B-chain (amino acids 25–54) that folds together with the A-chain (amino acids 90–110), both of which are connected by the C-chain (amino acids 57–87). The A- and B-chains have high conservation, whereas the C-chain is more divergent throughout vertebrate species (Figure 6A). Of the C-chain segments, only the known cleavage sites show high conservation (>95% across vertebrate species), suggesting that much of the folding dynamics of the proinsulin are driven by A- and B-chains, where the C-chain serves a role in tethering the intramolecular collapse. From NMR studies of proinsulin, model confidence suggests well-ordered A- and B-chains with less determinability of amino acid locations within the C-chain (Figure 6C), potentially indicative of C-chain disorder [158].

As with most proteins, hydrophobic collapse drives the initial seeding of the proinsulin structure through a complex, driven folding landscape [159,160]. Most importantly, several aromatic residues provide initial stable states for proinsulin folding [161]. These hydrophobic and aromatic residues within the core of the A- and B-chains include F49 and Y50 of the B-chain (B25 and B26 using chain numbers) and I91, V92, I99, L102, L105, and Y108 of the A-chain (A2, A3, A10, A13, A16, and A19). Most of these residues have vertebrate conservation >95% (Figure 6A). The F49L, V92L, and Y108C are known pathogenic or likely pathogenic variants for monogenic/neonatal diabetes [145,162,163,164,165] that fall in the core of mature insulin (Figure 6D).

### 2.8. Insulin Maturation, Storage, Secretion, and Receptor Activation Variants

Once the signal peptide has been cleaved within the ER, the protein continues folding into storage and secretion conformations. The folding of proinsulin appears to be a significant challenge for pancreatic beta-cells, with multiple environmental factors such as chronic ER stress influencing folding and resulting in the degradation of proinsulin [166,167,168]. One of the primary impairments of proinsulin folding is the formation of intramolecular disulfide alternative pairing, often through forming A6/A11 with A7/B7 mispairings, resulting in aggregated fibrils [169,170]. Thus, the regulation and formation of the intramolecular disulfide bonds are critical for insulin biosynthesis, yet the role of other missense variants altering this regulation is under-studied.

Folded proinsulin forms a noncovalent homodimer structure and is transported from the ER to the Golgi complex, where it is packaged into secretory granules for maturation [136,171]. The insulin dimer is stabilized in part by an aromatic triplet consisting of residues F48, F49, and Y50 (B24–B26) [172,173]. The homodimer utilizes amino acids F48, F49, and Y50 of the B-chain (B24–B26) [174], which forms a beta-sheet that hydrogen bonds between the monomers [175]. Two of these sites (F48, F49) are known to have variants associated with pathogenic monogenic diabetes (Figure 6A), with F48C being one of the three most common insulin variants in neonatal diabetes [145]. The Y50 (B26) was also shown to interact with the insulin receptor. Alteration of Y50 revealed that the conservation of the aromatic triplet is linked to dimer (and higher order hexamer) formation and receptor activation [172,176].

Following proinsulin transport to the Golgi complex, the C-peptide is cleaved to produce mature insulin. Not all proinsulin in the secretory granules is processed to remove the C-peptide, with multiple environmental factors such as inflammation influencing the balance of proinsulin to insulin [177,178,179]. Removal of the C-peptide within the Golgi complex and the secretory granules occurs at sites of basic amino acids (R55, R56, K88, R89) and involves different Ca-dependent endoproteases [180,181,182]. The cleavage of the B- to C-chains is more pH-sensitive than that of the C- to A-chain cleavage [183]. All four basic residues are highly conserved, with R55 and R89 having known pathogenic variants (Figure 6A) for familial hyper-insulinemia and neonatal diabetes [145,184,185]. The R89C is amongst the three most common causes of neonatal diabetes [145].

Insulin is organized into crystalline state regions within secretory granules, whereas the C-peptide is found in the surrounding fluid [186]. The ZnT8 and SLC30A8 transporters found within the granules are involved in the transport and accumulation of intra-granule Zn [187]. This high Zn concertation transforms the insulin dimers into hexamer structures through H34 (B10) histidine coordination (Figure 7), resulting in the crystalline state [175]. Hexamers are formed from a trimer of dimers coordinated around two Zn^+2^ ions (Figure 7A,B) [188]. The additional presence of other molecules, such as chloride and phenol, can impact the allosteric structure of the hexamer [189]. The mapping of variants to the hexamer structure (Figure 7C) [190] showed the H34 (B10) amino acid variant as well as H34D, which is associated with neonatal diabetes [163,165], to fall on the Zn binding site (Figure 7D). Additional residues with known disease variants that fall within hexamer structure contacts (Figure 7E–G) include H29, L30, G47, F48, F49, Y50, and P52 (B5, B6, B23, B24, B25, B26, and B28). Following the release within the blood, the Zn level is far lower and prevents the continued hexamer complex.

The around three-hour maturation process [180] of insulin containing secretory granules finishes with docking near the plasma membrane [191]. The beta-cells are stimulated for granule release by various dietary components, notably glucose, with multiple components of the pathway possible targets of the genetic etiology of diabetes [192]. Secretion is altered by amino acid, lipid, and incretin hormones [193]. Insulin-containing granule release also results in the release of other endocrine regulators, including the processed C-peptide, amyloid polypeptide, IGF2, secretogranins, GABA, and ATP [194].

Circulating insulin is found as a monomer without known carrier proteins. Unlike IGF-1, which is regulated in the blood by IGFBPs, insulin diverges in sequence to not interact with these circulating carrier proteins [195,196]. Once circulating, insulin can bind to the insulin receptor expressed in nearly every tissue and cell (Figure 4A). Similar to the insulin structure, the hydrophobic and beta-sheet segments are involved in receptor binding [197]; however, multiple conformations of binding were observed in two different receptor binding sites [198,199,200,201] (Figure 8). Recent advances in cryo-EM and crystallography have allowed the nearly complete human insulin receptor structure to be solved with insulin binding, as seen from the BLAST of the PDB (Figure 8A). The dimeric insulin receptor, upon binding, undergoes an allosteric alteration to activate the intracellular tyrosine kinase [202]. Mapping the variants from our screen to the binding sites of two different solved full receptor and insulin interactions (Figure 8B) showed potential for diabetes-associated variants at amino acids H29, L30, H34, L35, L39, V42, G44, R46, G47, F48, F49, Y50, V92, E93, T97, S98, S101, Y103, and Y108 (Figure 8C,D). Variation within the insulin receptor and autoantibodies to the receptor can also result in insulin resistance [203,204,205,206], requiring further work from the receptor perspective.

Insulin is cleared from the blood through receptor binding, liver degradation, and kidney filtration [207,208]. The hepatic pathway clears an estimated 50–80% of circulating insulin with enzymes such as CEACAM1 [209,210]. Within the kidney, following glucose-stimulated elevation, insulin can be filtered at higher rates with increased urine abundance [211]. The exogenous delivery of insulin also results in increased urine protein levels [212]. Alterations to many of these clearance pathways can impact diabetic risk [213,214,215,216]. Within the kidney, there is a high correlation between insulin resistance and glomerular filtration rates during chronic kidney disease [217,218]. The role of insulin variants on clearance pathways has not been explored in depth.

C-peptide is equally secreted with insulin [219]. Secreted C-peptide was shown to possess neuroprotective [220] and renal-protective functions [221], suggesting that it is not inert but has endocrine properties [222]. The C-peptide has an insulin-independent activation of surface G-protein-coupled receptors [223]. Exogenous delivery of proinsulin has some elevated hepatic localization and adverse clinical effects [224]. While the C-chain displays weak selection (Figure 6A), several amino acids are highly conserved, including E59, L68, G75, L77, and E83. These amino acids could be critical to proinsulin processing and folding [225], or they could be involved in some unknown role of secreted C-peptide. The G75D variant (rs144093133) is the most common gnomAD missense variant within the *INS* gene to have predicted detrimental function (Figure 3H). It results in the addition of polar acidic amino acid, is found to be heterozygous in 0.2% of African/African American individuals, and has an odds ratio of 17 for type 2 diabetes [226], suggesting variants within the C-peptide are vastly under-studied.

As insulin is very difficult to measure in the blood, the C-peptide is commonly used to measure, as a proxy, blood insulin levels [227]. Measurement of the C-peptide has shown utility in understanding the clinical progression of diabetes and measuring endogenous insulin production in patients with exogenous insulin treatment [228,229]. As the C-peptide is usually measured through immunoreactivity or mass spectrometry [230,231,232], amino acid changes could impede accurate measurements. The change of the inert glycine at amino acid 75 to the acidic aspartic acid should significantly impact the antibody binding of immunoreactivity while also resulting in peak shifting in mass spec. The diversity of antibody sources is highly correlated with the diversity of C-peptide measurements [233]. We speculate that rare variants of the C-peptide remain one of the most underappreciated components of insulin biology.

Generally, in the healthy state of pancreatic beta cells, the maturation of proinsulin is complete before secretion. In some states of disease and beta-cell dysfunction, proinsulin secretion increases [234]. Proinsulin has a significantly longer half-life in blood relative to insulin due to decreased hepatic clearance [235,236]. Like insulin, proinsulin can also bind to the insulin receptor. However, in healthy states, it represents a small portion of the ligand–receptor complex [237] and binds with a 100-fold lower affinity [238,239]. However, the alternatively spliced insulin receptor found in fetal tissue and cancer cells has a higher affinity for proinsulin [240]. This receptor isoform is similarly bound by IGF-2 [241] and involved in embryonic regulation [242], which suggests the secreted proinsulin may play a role in development, such as that of the pancreas. When considered with this view, a seemingly insignificant C-chain mutation could result in a differentiation trajectory change during embryonic development, potentially causing a predisposition to disease before an individual is born.

### 2.9. Animal Models and Insulin Gene Duplication

Model organisms have provided insights into insulin biology and diabetes, with multiple reviews published on this topic [243,244,245,246]. From a genomics perspective of these models, the genome duplication of *insulin* within mouse, rat, and zebrafish is striking. Most functional amino acids with variants discussed are found within both insulin copies of these species (Figure 9). Within mice, the knockout of either *insulin* gene (*Ins1*-NP_032412, *Ins2*-NP_032413) can influence signaling and diabetic progression [247], with the *Ins2* gene knockout more critical for age-dependent hyper insulinemia and T1D [248]. Within the rat, the two *insulin* copies (*Ins1*-NP_062002, *Ins2*-NP_062003) have similar functions [249], but differences in the 5′ flanking sequence impact expression of the two genes [250]. Within zebrafish, the two insulin copies (*ins*-NP_571131, *insb*-NP_001034153) were suggested to play a role in glucose homeostasis, with *ins* found expressed within the pancreas and *insb* more broadly [251]. It should be noted that the hexamer important to B10 histidine is changed to a serine in the protein coded by *insb* (green, Figure 9), suggesting it may be more divergent in pancreatic function than *ins*.

### 2.10. INS–IGF2 Protein Sequence

The *INS* gene is located on the same DNA strand as *IGF2,* with potential readthrough and splicing joining the two into the *INS–IGF2* transcript. Within metabolic endocrinology, *INS–IGF2* is not the only known readthrough event. The Endospanin coding gene has readthrough for the leptin receptor, which enhances and broadens the leptin receptor expression profile [252]. While some groups have suggested the readthrough transcript of *INS–IGF2* to be an “artifact” with 20,000 times lower expression in beta cells than *INS* [253], there is strong support that the readthrough is present and functional within the pancreas. Deep human resequencing of RNA has elucidated the transcript to be present in the placenta, liver, pancreas, fat, and ovary [254]. Within the pancreas, targeted analysis shows the persistence of *INS–IGF2* even when the *INS* promoter is inactive [255]. Within pediatric T1D, serum autoantibodies show elevated INS–IGF2 immunoreactivity relative to controls [256]. Terrestrial vertebrates share synteny between the two genes [257]. In rats and mice, *Ins2* is syntenic with *Igf2* and located proximally to each other, with proteomics revealing rodent fusion proteins [258]. Within the pig, the intergenic segment of these two genes more similarly reflects the human sequence than the rodent [259]. The *INS–IGF2* readthrough transcript is annotated in 10 additional species (*Ursus americanus*-LOC123775522, *Puma concolor*-LOC112852063, *Camelus dromedarius*-LOC105106405, *Rattus norvegicus*-LOC102548568, *Ictidomys tridecemlineatus*-LOC120889510, *Bison bison bison*-LOC104987751, *Lynx canadensis*-LOC115526480, *Trichechus manatus latirostris*-LOC105755980, *Camelus ferus*-LOC106729876, and *Arvicola amphibius*-LOC119800954).

The two human *INS–IGF2* transcripts code for the same protein (Figure 10). The resulting protein has a conservation of the insulin signal peptide, B-chain, and the cleavage site between the B-chain and C-chain (Figure 10A). Within the first several amino acids of the C-chain is the alternatively used splice site that creates a unique sequence from the IGF2 5′UTR. Taking 20 bases before and after the splice site nucleotides (underlined, Figure 10A), this sequence is highly unique to humans *INS–IGF2* based on BLAST to the Nucleotide Collection (Figure 10B). This uniqueness allows for the use of the sequence within human RNAseq extraction insights, as no other human transcript matches it with this level of statistical cutoffs. We show that samples annotated with high levels of ENST00000397270 based on quasi-alignment have high levels of reads unique to this splice site (Figure 10C). However, samples with elevated NMD-regulated ENST00000356578 tend to have lower levels of unique splice site reads, suggesting that the extended UTR of the transcript enhances the TPM mapping within quasi-aligners. Few pancreatic long-read datasets (PacBio or Nanopore) are available as of 2022, which will be a critical technique in characterizing fusion transcripts such as *INS–IGF2*.

From the protein sequence, it should be noted that following the splice site between *INS* and *IGF2*, there is a large ORF (Figure 10A). This sequence is out of frame from the normal IGF2 ORF, with most of the sequence found within the *IGF2* 5′UTR. This sequence has no known protein structures or homology to other human proteins. The presence of multiple polar basic, ER-regulated cleavage sites within the IGF2 portion is striking, suggesting cleavage products of the protein if made in the rough ER. As most 5′UTRs are selected for lack of an open reading frame, the remaining protein size seems to be selected to avoid an earlier stop codon.

With so many human genomes sequenced, there would be a high probability of observing missense and nonsense variants within the gene if the fusion protein was nonfunctional. However, according to gnomAD, the frequency of missense variants is 0.01 standard deviations relative to all other genes of the genome, and that of nonsense variants is 0.06, suggesting there is a lack of negative selective pressure within human genomes that maintains the function of the ORF. Only eight ultra-rare (observed 1–2 individuals) frameshift or nonsense variants are seen from 76,125 whole genome sequences, with 6/8 of them located in the IGF2 region (Figure 10D). Of the missense variants, most are also rare. The Leu144Pro change (rs10770125) is found in 62% of Amish and 56% of Finnish European ancestry individuals, whereas only 18% of African American ancestry individuals have the variant. Its association with T1D (https://genetics.opentargets.org/variant/11_2147784_A_G, accessed on 6 December 2022) is likely due to its LD with rs689 (0.22 R^2^). All other missense variants are below allele frequency for GWAS and, thus, not associated with any known traits. Interestingly, the number of variants per base within the *INS* and the *IGF2* region are not that different (Figure 10E), suggesting that the pressure to maintain insulin amino acids is very similar to the *IGF2*-coded region of INS–IGF2 protein.

The most surprising variant observed in gnomAD within 13 individuals was the loss of the stop codon for INS–IGF2. This variant results in a stop codon change to glycine resulting in 28 additional amino acids (p.Ter201GlyextTer29, rs769367976). The change occurred in 11 African/African American individuals (Figure 10F), suggesting a major under-explored potential variant of INS–IGF2.

The most challenging area of insulin genetics is understanding noncoding variants that might influence *INS* readthrough potential. Imprinting is critical to this *INS* and *IGF2* region [260,261,262], with methylation alterations in different cell types [263]. It was suggested that this fusion transcript can have oncogenic properties and is expressed in various cancers [44,264]. This suggests that readthrough and transcript fusion may have an inheritance or environmental regulation. The polymerase must continue unopposed on the DNA for readthrough between the two genes. This means various insulators and transcriptional repression complexes must not be present. How rare variants impact readthrough has never been explored. We show that several regulation sites between *INS* and *IGF2* (Figure 3F,G) have potential variants that influence transcription factor binding. Thus, we anticipate that noncoding variants influencing readthrough may be a promising new avenue for diabetic and metabolic research.

## 3. Discussion and Future Directions

The American Diabetes Association (ADA) in its consensus report on developing precision medicine guidelines identified being able to define discrete subgroups within diabetes as crucial to developing insights into etiology, prognosis, planning treatment, and monitoring response [265]. As a greater number of individuals are sequenced, more pathogenic variants will be identified. Because next-generation genome sequencing is getting cheaper and faster, there is increased funding from hospitals, coverage from insurance companies, and research organization investments to expand into more diverse individuals. Metabolic health will likely become a significant initiative for understanding genomic etiology by integrating everyone’s genetics with other clinical metrics. Many lessons learned in this review for insulin will apply to other metabolic genes. The role of rare noncoding or transcript processing site (splice and UTR) variants remains one of the most challenging aspects of insulin biology from genome sequencing. Divergent splicing and the resulting proteins remain underexplored, as highlighted here for the importance of *INS–IGF2* transcripts in diabetes. Further investments in RNAseq and, more importantly, long-read sequencing strategies (such as PacBio and Nanopore) will continue advancing our knowledge of metabolic transcripts and their expression changes in disease states.

Next, there must be additional investments into understanding the epigenetics of metabolic tissues such as pancreatic islet cells, something we showed is incredibly lacking relative to many other cell types. Advances in the understanding of pancreatic beta cell biology will also lead to the treatment of disease in the future. New tools such as induced pluripotent stem cells might enhance our ability to generate these datasets in healthy and diseased states. Stem cell replacement therapies of beta cells hold promise as a potential cure, but significant technical problems still prevent this treatment from being used clinically [266]. Integrating diverse datasets is necessary to develop a more robust working knowledge of how genetics can influence insulin biology and to effectively understand an individual’s risk of developing diabetes and diabetic complications. As we showed within this article through a review of literature and data, much work is still to be done in defining genetics for one of the most well-studied human genes. These future research needs of insulin reflect the current limitations for nearly every human gene, serving as an example to guide future genetic investments.

The genetics of insulin shows highly penetrant rare variants, common high-risk alleles with lower penetrance, and what appear to be genetic modulators influenced by environmental factors. Variants within one gene can result in monogenic/neonatal diabetes, T1D, insulin resistance, T2D, hepatic diabetes, or glucose signaling alterations. The plasticity of diabetes and diabetic outcomes has remained a critical challenge in endocrinology. The regulation of *INS* transcription, transcript splicing, transcript regulation (especially by variable UTR regions), translation of preproinsulin, processing/maturation of proinsulin, secretion, receptor binding, and metabolism can all be modified to impact insulin biology (Figure 11). Direct *INS* variants can differentially influence various portions of these pathways, resulting in diverse phenotypes depending on where/how the variant changes molecular biology.

The rs689 variant confers the most significant *p*-value of all T1D, yet it only has an odds ratio of 1.5–2.3. Thus, it is found throughout a large portion of the population without disease. This suggests either polygenic risk (the interaction of multiple independent risk variants) or that something from the environment influences this variant’s mechanism. Multiple other loci confer similar odds ratios for T1D and are from various pathways, suggesting that some polygenic insights are necessary to understand insulin within the complex T1D genetics. Yet, environmental factors may hold more insights into why many variants have low penetrance for T1D. As our RNAseq analyses have shown, there is a promise for understanding how components such as viral infections or cellular/ER stress might interact with risk alleles such as rs689 to result in T1D development.

Understanding the entire process of insulin biosynthesis and the many components involved will be crucial in future diabetic research. While this article focused on the locations of current known variants within solved protein structures, a wealth of knowledge of the biophysics of insulin dynamics needs additional integration into the larger variant potential for interpretations. The role of genetic variants in any of the insulin biosynthesis steps could modulate the risk of diabetes. Many transcription factors can impact insulin transcriptional regulation, whereas transcripts are regulated by splicing factors, NMD pathways, and mRNA processing from the nucleus to ribosomes (Figure 11A). The synthesis of proinsulin requires translation initiation machinery, folding chaperones, and multiple ER components (Figure 11B). The folding and storage of insulin require enzymes, Zn transporters, and secretory vesicle machinery (Figure 11C). Insulin release and bioactivity involve multiple cellular receptors, secretory vesicle exocytosis machinery, insulin receptor downstream signaling, and insulin clearance pathways (Figure 11D). The receptor interaction and metabolism pathways represent one of the most exciting aspects of insulin precision medicine, as it may hold additional answers for therapeutic derivative choices within each individual. Nearly every one of the steps within Figure 11 is error-prone with the potential for genetic and environmental interactions to impact insulin synthesis. Similar to our literature and data review performed here for insulin, every one of the proteins involved in insulin biology could benefit from a detailed exploration of genetic variants and disease involvement.

Genetics alone are not the only way of modifying these complex steps of insulin biology, but multiple factors of the environment can also impact these pathways (Figure 11E). One of the environmental components of insulin that is well-studied is the role of ER stress [267,268,269,270,271]. Chronic ER stress caused by the increased need for insulin synthesis can potentially cause beta cell de-differentiation and death. Genes linked to ER membrane homeostasis, such as *WFS1,* were connected to maturity-onset diabetes of the young [272]. Genetic variants were shown to result in the development of ER stress-induced diabetes in mice when exposed to a high-fat diet [273], suggesting that genetics and environment can interact with ER stress and diabetes risk.

Many other environmental factors can influence insulin biology (Figure 11E). While it is largely unknown why, the immune system can develop antibodies to insulin, which is potentially connected to some unknown regulation of *INS–IGF2* [44,274,275,276]. Transcriptional alterations through factors such as TCF1, TCF2, HNF4A, IPF4, NEUROD1, and TCF7L2 were linked to various types of diabetes, with several directly connected to altering insulin biosynthesis [277,278,279]. As discussed in Section 2.4, multiple viruses can increase the risk of diabetes, with a need to further understand direct viral regulation on *INS–IGF2* NMD-regulated transcripts and ER stress. While we provided examples of a few environmental factors that can regulate insulin, there are likely many other mechanisms. As we move forward, it will be essential to build these environmental factors into our genetic modeling of insulin biology and diabetic risks.

Pancreatic beta cell biology is incredibly complex with thousands of molecular interactions and pathways. Genetic risk can cause disease in any of these pathways; however, that risk might not become activated unless an environmental factor interacts with these genetics. As we have demonstrated in previous sections with the analysis of public pancreatic RNAseq datasets, there is a need for increased research into the role of the environment on transcription, splicing, and protein production. Because there are many possible variables in population environmental studies, big data research methods should be used to integrate genetic risk through GWAS and RNAseq analyses. With these newer big data strategies, it will be possible to develop our knowledge of mechanisms for diabetic genetics through environmental interactions that have been elusive until now.

## 4. Methods

All genome browser data was extracted from the UCSC Genome Browser [280] on 1 November 2022. On the same day, the GWAS traits were extracted from the GWAS catalog [50], the top SNP traits for INS variants were extracted from Open Targets Genetics [281], and variant population frequencies were extracted from dbSNP [62]. Linkage disequilibria for top variants were extracted using the SNiPA tool [282]. All extracted variants were processed through the Ensembl Variant Effect Predictor (VEP) [88] on 8 November 2022 with all possible tools turned on. Noncoding variants were run through the RegulomeDB tool for gene regulation insights [92]. Nucleotide sequences of *INS* were extracted from the NCBI ortholog [283] and aligned using Mega tools [284] with ClustalW [285]. Amino Acid sequences were aligned using COBALT [286].

GTEx expression data [61] of *INS*, *INS–IGF2*, and *INSR* were extracted from the transcript per million (TPM) version 8 file (https://storage.googleapis.com/gtex_analysis_v8/rna_seq_data/GTEx_Analysis_2017-06-05_v8_RNASeQCv1.1.9_gene_tpm.gct.gz, accessed on 6 December 2022). FASTQ files from the NCBI SRA for pancreas or liver were downloaded using the SRA toolkit [287]. The files were aligned to human Gencode 42 transcriptome [288] using Salmon [289]. Heatmaps were generated using the Broad Institutes Morpheus tool [290].

All protein coordinate files were sourced from the RCSB’s protein data bank [291]. Molecular graphics and analyses (hydrogen bond approximation and electrostatic surface coloring) were performed with UCSF ChimeraX, developed by the Resource for Biocomputing, Visualization, and Informatics at the University of California, San Francisco [292]. Figure 8A,B interaction color groups are based on known structural interactions from the extended data of Table 1 of [293], of which the information was originally published in [201].

## Figures and Tables

**Figure 1 biomolecules-13-00257-f001:**
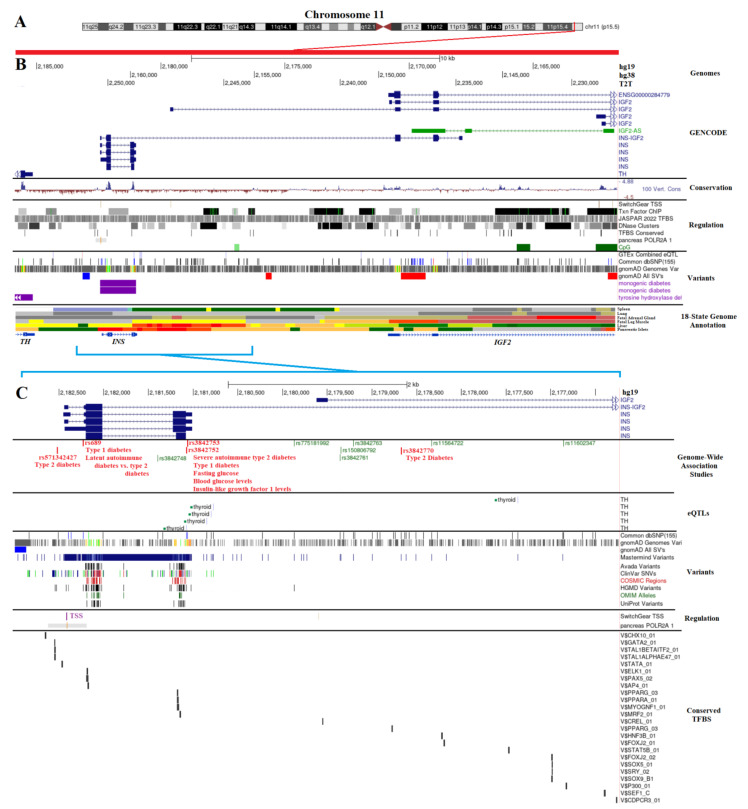
Genomic region of insulin. (**A**) The map of chromosome 11 with the *Insulin* (*INS*) gene region in red on the chr1(p15.5) segment. (**B**) Genome browser annotation (UCSC, https://genome.ucsc.edu/ (accessed on 6 December 2022) for a region near *INS* (hg19-chr11:2,161,620–2,185,936, hg38-chr11:2,140,390–2,164,706, T2T-chr11:2,228,043–2,254,072). This covers the region between *Tyrosine Hydroxylase* (*TH*) and *Insulin-like growth factor 2* (*IGF2*). At the top are the chromosome labels for three different human genome releases (hg19, hg38, and T2T). Below are known splicing isoforms of GENCODE, conservation (blue peaks represent conserved regions), various regulation datasets, known human variants, and the Roadmap Epigenomics 18-state genome annotation of several tissues, with pancreatic islets shown at the bottom. (**C**) Zoomed-in view of *INS* marked in cyan on Panel B (hg19-chr11:2,176,237–2,182,988, hg38-chr11:2,155,007–2,161,758, T2T-chr11:2,242,655–2,251,112). Shown are known Genome-Wide Association Study (GWAS) loci for biological traits with diabetes-connected SNPs in red, known expression quantitative trait loci (eQTLs) from GTEx, known human variants from various databases, known regulation sites, and conserved transcription factor binding sites (TFBS). All panel colors from the UCSC genome browser extractions can be found within track information within the browser, as we have used default color schemes from UCSC.

**Figure 2 biomolecules-13-00257-f002:**
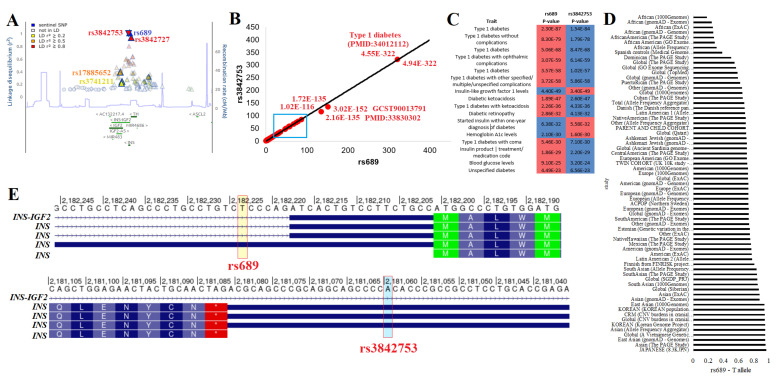
Top LD block for *insulin* and diabetes. (**A**) Linkage disequilibrium map of rs689, with other T1D associated SNPs labeled. Those in red have >0.8 R^2^ with rs689, orange, >0.5 R^2^, and yellow, >0.2 R^2^. Shown below is the location of genes with green lines representing the gene location. The blue line represents the recombination rates across the region. (**B**) Open Target Genetics extracted −log10 *p*-values from various associations for rs689 (x-axis) relative to rs3842753. The black line shows a 1-to-1 relationship. (**C**) *p*-values are shown for various traits in the Panel B cyan box for both SNPs. The more significant values are shown in red. (**D**) The rs689 T allele frequency from various genomic datasets ranked from lowest to most frequent. (**E**) The genomic location (hg19) for rs689 and rs3842753 relative to transcripts of *INS*. Amino acids are labeled in boxes, the UTRs are represented as blue lines, and the intron segments are represented with arrowed lines.

**Figure 3 biomolecules-13-00257-f003:**
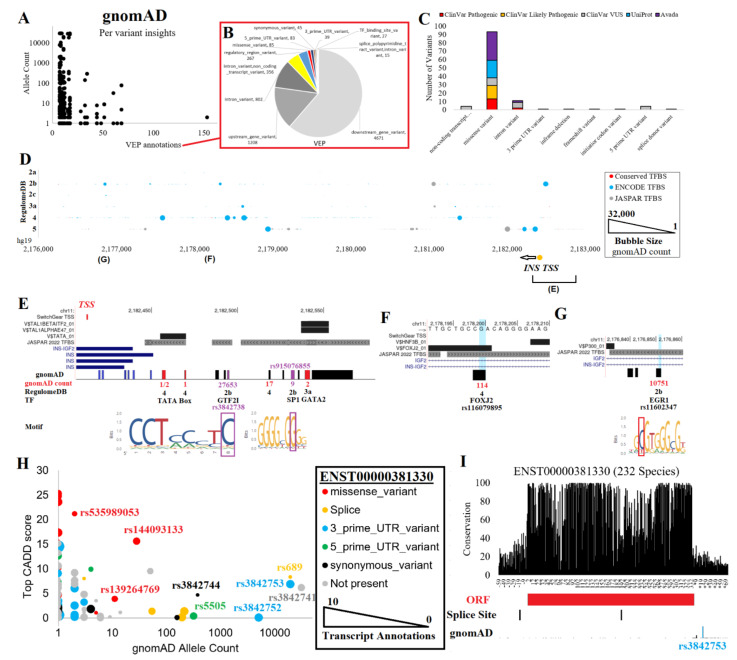
*INS* variants. (**A**) gnomAD variants for the region shown in Figure 1C. The x-axis shows the number of VEP annotations for each unique variant, and the y-axis shows the gnomAD allele count. (**B**) The red box calls out the breakdown of VEP annotations over all variants. (**C**) Unique variant annotations for ClinVar, UniProt, and Avada. (**D**) Noncoding variant annotations. The x-axis shows the genome coordinates (hg19), the y-axis shows the RegulomeDB score, and the bubble size represents the gnomAD count. Red dots denote those within a conserved transcription factor binding site (TFBS), cyan dots for those within a known ENCODE ChIP-Seq binding event, and gray dots for those within a JASPAR-predicted TFBS. Below is the INS transcriptional start site (TSS) and two regions covered in Panels E-G. (**E**) Annotation of variants within the promoter of *INS*. The TSS is shown in red. Conserved TFBS and JASPAR are shown above *INS* transcripts. On the bottom are the gnomAD variant counts for RegulomeDB predictions or for those that fall within a conserved TFBS. (**F**) A downstream enhancer site with rs116079895 that is predicted to alter a FOXJ2 conserved TFBS. (**G**) A downstream enhancer site with rs11602347 predicted using RegulomeDB to alter an EGR1 binding site. (**H**) Transcript annotations from VEP. Colors represent the labeled ENST00000381330 annotations, the primary *INS* transcript. The *INS* transcript annotations found outside of ENST00000381330 are in gray. The x-axis shows the gnomAD allele count, the y-axis shows the top CADD score for functionality, and the bubble size shows the number of transcript annotations for each variant. (**I**) Conservation of 232 species transcripts for ENST00000381330 at the RNA base level. Below are the open reading frame (ORF), transcript splice sites, and gnomAD allele counts.

**Figure 4 biomolecules-13-00257-f004:**
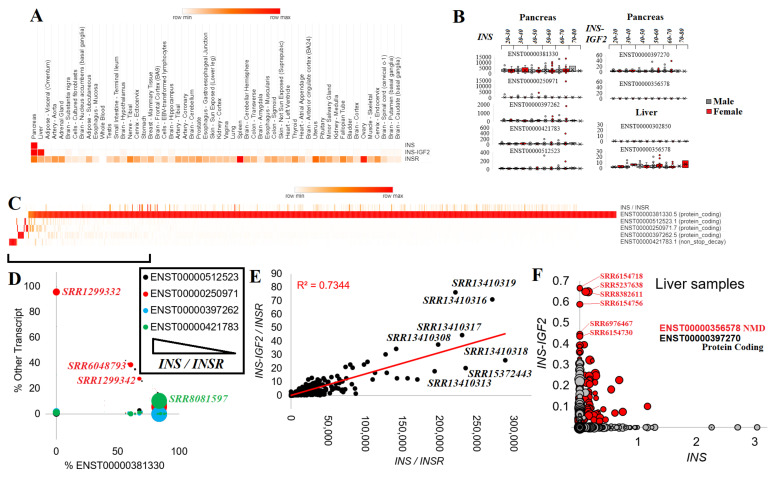
Expression of *Insulin* (*INS*) in public datasets. (**A**) GTEx average expression of *INS*, *INS–IGF2*, and *INSR* over various tissues. The heat map shows the max expression of each gene in red. (**B**) Box and whisker plots for the various *INS* (left) or *INS–IGF2* (right) transcripts in the pancreas and liver relative to each age group of samples. Male samples are shown in gray, and female samples are shown in red. (**C**) Heat map of *INS*/*INSR* ratio and each of the five transcripts of *INS* for 1360 “Pancreas”, “Islets”, or “Langerhans” human RNAseq samples. Values for *INS* represent the percent of total *INS* levels for each transcript. (**D**) Individual sample breakdown from Panel C for those with <90% of the transcript abundance for ENST00000381330 (x-axis) shown relative to the other transcripts. The size of the bubble represents the total INS/INSR levels. (**E**) Normalized values for *INS* (x-axis) and *INS–IGF2* (y-axis) relative to *INSR* for various pancreas RNAseq BioProjects. The red line is the best-fit trendline with the R^2^ shown. (**F**) The expression of *INS* and *INS*-*IFG2* in transcripts per million from 1901 “liver” or “hepatocyte” human RNAseq samples. The red transcripts are those of *INS–IGF2* ENST00000356578. The size of the bubble represents the expression of *Albumin* (*ALB*), a marker of hepatocytes.

**Figure 5 biomolecules-13-00257-f005:**
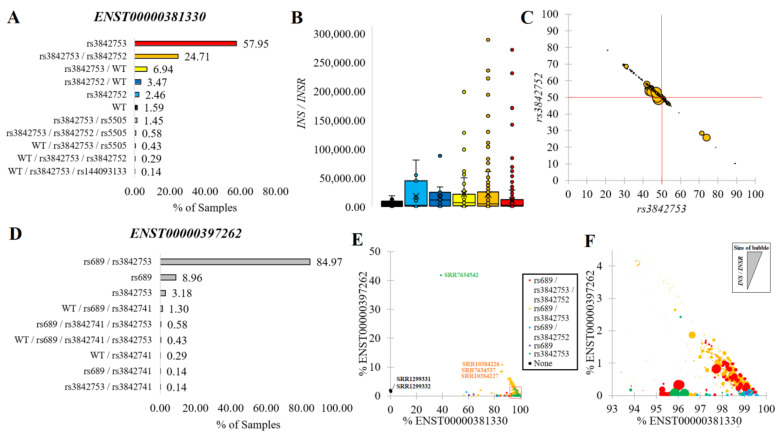
*INS* variants within transcripts. (**A**) The percentage of samples with transcripts annotated with unique and combinations of variants within ENST00000381330. (**B**) Box and whisker plots for *INS*/*INSR* normalized levels for groups as colored in Panel A. (**C**) The percent of ENST00000381330 reads containing rs3842753 (x-axis) relative to rs3842752 (y-axis) for compound heterozygous samples of rs3842753/rs3842752. The size of the bubble represents the *INS*/*INSR* normalized value. The red lines are at the 50% level. (**D**) The percentage of samples with transcripts annotated combination of variants for ENST00000397262. (**E**) Combination of Panels (**A**) and (**D**) for labeling each sample’s final genotype (colors) relative to the % of total transcripts coming from ENST00000381330 (x-axis) and relative to ENST00000397262 (y-axis). (**F**) Zoomed-in view of the Panel E red box with the bubble size representing the *INS*/*INSR* normalized value.

**Figure 6 biomolecules-13-00257-f006:**
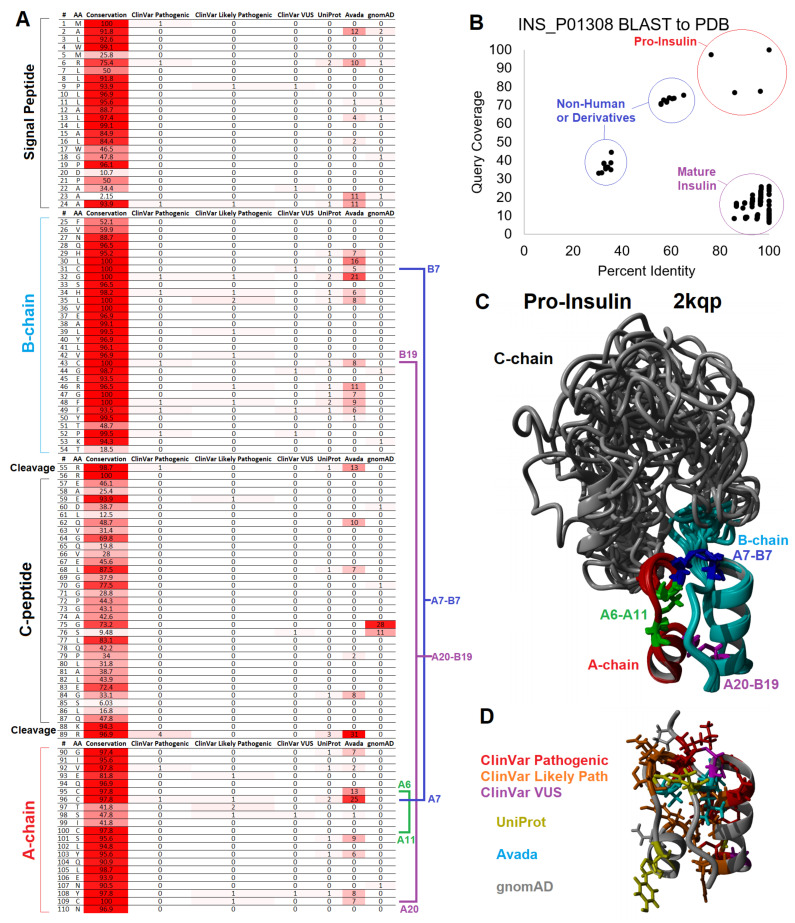
Mapping of variants to insulin structure. (**A**) Table of conservation and missense variants from ClinVar, UniProt, Avada, and gnomAD impacting the preproinsulin molecule listed by amino acid. Cystines involved in disulfide bonding are marked in blue (A7-B7), magenta (A20-B19), and green (A6-A11) with labeling for A- and B-chain numbering. (**B**) BLAST analysis of sequences within the PDB similar to human preproinsulin (UniProt O01308) with the percent identity of the sequence from the structure shown on the x-axis and the coverage of sequence on the y-axis. (**C**) Structure models of human proinsulin from PDB file 2kqp showing the aligned top twenty models. The B-chain is colored cyan, C-chain in gray, A-chain in red, and disulfide bonds as colored in Panel A. (**D**) The A- and B-chain are the same orientation as Panel C but colored for human variants, where red represents ClinVar variants identified as pathogenic, orange represents ClinVar variants identified as likely pathogenic, magenta represents ClinVar variants of uncertain significance (VUS), yellow represents UniProt-annotated variants, cyan represents Avada-annotated variants, and gray represents gnomAD-observed variants.

**Figure 7 biomolecules-13-00257-f007:**
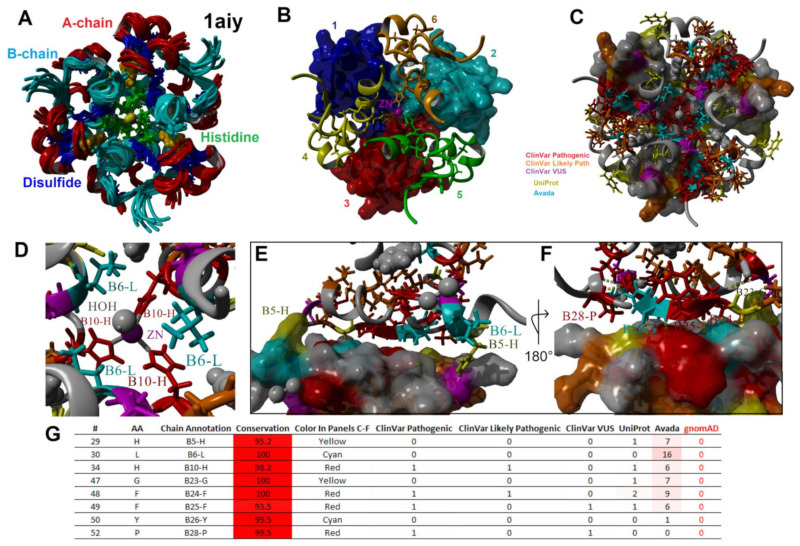
Mapping variants to the multimeric state of insulin. (**A**–**C**) The PDB file 1aiy shown in the same orientation but colored various ways. Panel A shows the A-chain in red, B-chain in cyan, disulfide bonds in blue, histidine in green, and zinc in magenta for the top ten predicted models of the NMR structure. Panel B shows the top model of Panel A for the six different insulin molecules (both A- and B-chains) in different colors, with the first zinc coordination site molecules shown as surface plots and the other zinc sites as ribbon plots. Panel C uses the same coloring as Figure 6D for human variants. (**D**) A zoomed view of one Zn coordination site using the coloring of Panel C for human variants. Labeled residues are those that make contact between multiple subunits. (**E**,**F**) Zoomed view of contact residues between different insulin molecules. Labeled amino acids have a human variant. Panel F is a 180-degree rotation of Panel E. (**G**) Table of missense variants forming contacts within the hexamer subunits.

**Figure 8 biomolecules-13-00257-f008:**
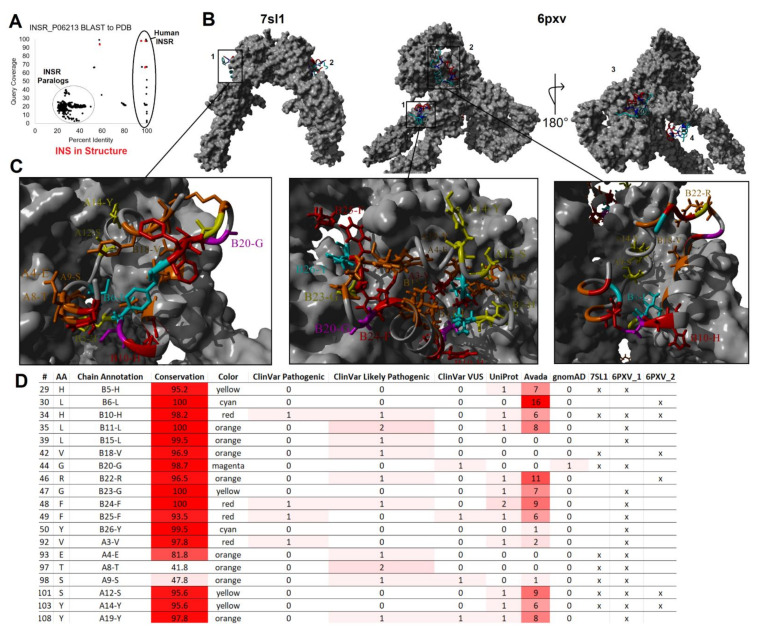
Mapping variants to insulin and insulin receptor interaction. (**A**) BLAST analysis of sequences within the PDB similar to human insulin receptor (UniProt P06213) with the percent identity of the sequence from the structure shown on the x-axis and the coverage of sequence on the y-axis. Spots in red are present in Figure 6B, meaning they represent insulin and insulin receptor interactions. (**B**) The model of the full insulin receptor (gray) from PBD files 7sl1 and 6pxv. Insulin is colored with A-chain in red and B-chain in cyan with two insulin bound for 7sl1 and four for 6pxv. (**C**) Zoomed-in view of insulin docking with the receptor as called out in Panel B. Coloring of insulin is based on Figure 6D. (**D**) Table of insulin missense variants that form contacts from one of the three binding states of Panel C. The last three columns label the structures that contacts are found for insulin receptor with an x.

**Figure 9 biomolecules-13-00257-f009:**
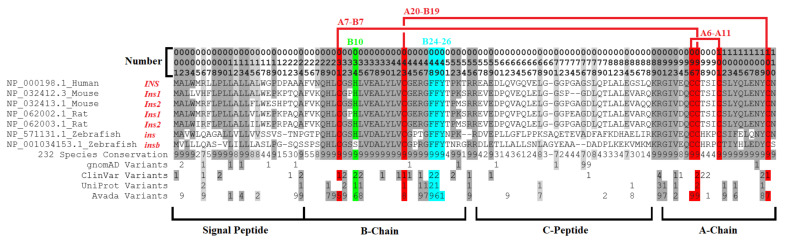
Compiled insulin alignment data including duplication copies of mouse, rat, and zebrafish. The numbers listed at the top can be read as the first digit representing the hundreds place, followed by the tens, and the last number as the single digit. For conservation and all listed human variants, a value of nine is the highest, representing anything ≥9. The conservation is shown as the percent of 232 vertebrate species, where 9 = 100–90% conserved, 9 = 80–90%, and so on. Conserved sites at 7 or higher are gray, with those 9 in darker gray. Red are disulfide bonds, cyan is the dimer region, and green is the hexamer site.

**Figure 10 biomolecules-13-00257-f010:**
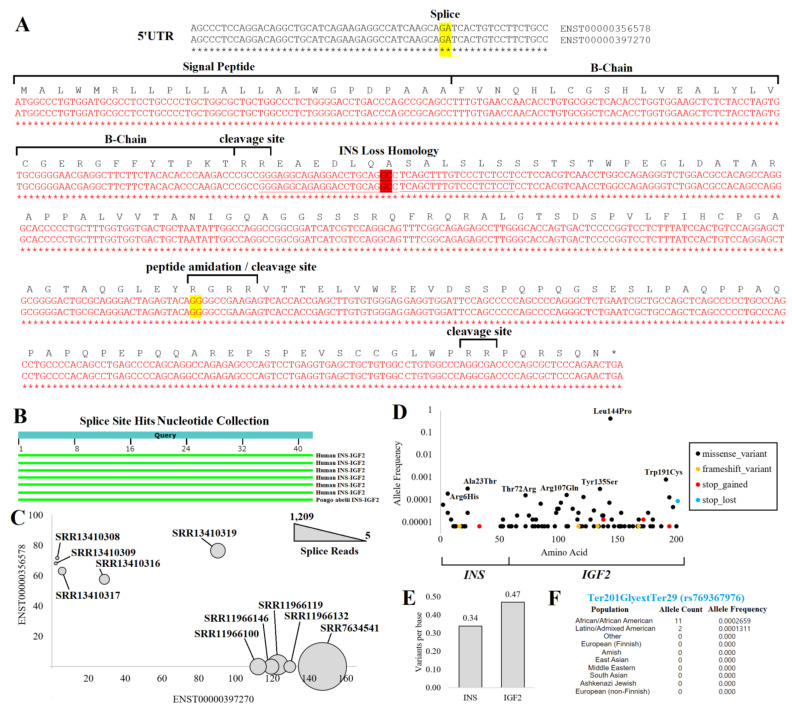
INS–IGF2 protein insights. (**A**) Nucleotide to amino acid alignment for the two INS–IGF2 transcripts. Splice sites are highlighted in yellow, with the fusion splice site highlighted in red. The underlined sequence was used for the unique detection of fusion splicing. The nucleotide sequence on top is from ENST00000356578. and the bottom is from ENST00000397270. The INS Loss Homology site is where the INS–IGF2 diverges from INS sequence. (**B**) Nucleotide collection BLAST hits of the underlined sequence for unique fusion splicing. (**C**) SRA BLAST hits of fusion splice sites (bubble size) for ten RNAseq samples relative to Salmon-based annotations of each *INS–IGF2* transcript. (**D**) The gnomAD version 3 variants of the fusion transcript show the allele frequency (y-axis) relative to the amino acid location (x-axis) for variants that influence the protein. (**E**) The probability of unique variants per base found within the protein’s INS or the IGF2 region. (**F**) Allele frequencies from various populations for the INS–IGF2 loss of stop codon variant.

**Figure 11 biomolecules-13-00257-f011:**
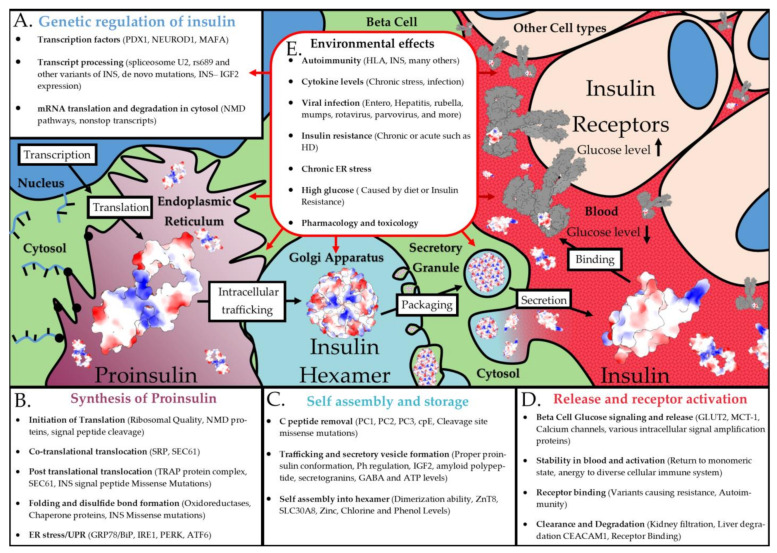
Environmental effects on the synthesis of insulin. Shown are factors involved in *INS* transcriptional regulation (**A**), proinsulin synthesis (**B**), insulin assembly/storage (**C**), and insulin release/receptor activation (**D**). This is shown overtop of an artistic rendering of the complex insulin biosynthesis. (**E**) Environmental factors that can potentially impact every step described in the prior panels.

**Table 1 biomolecules-13-00257-t001:** BioProjects with *INS* expression from pancreas.

BioProject	Details	Samples with *INS*	*INS* TPM	*INS–IGF2* TPM	*INSR* TPM	% rs3842753	% rs3842752	% rs689
PRJNA217347	pancreatic islet donors	86	16,730.85	4.62	15.73	58.70	33.41	58.02
PRJNA587101	circadian control of human in vitro islet maturation	84	9810.66	0.93	14.03	99.22	0.00	21.76
PRJNA422401	pancreatic islet donors	82	158,258.15	14.55	11.46	79.46	14.37	73.37
PRJNA638360	human islets	66	244,231.55	63.57	16.38	71.44	21.84	65.93
PRJNA671458	human β cell dysfunction	64	120,539.80	18.46	13.60	61.94	22.69	85.29
PRJNA723013	pancreatic adenocarcinoma	38	690.01	0.03	15.50	96.34	3.49	44.59
PRJNA865345	pancreatic cancer precursors	23	3871.01	0.22	18.19	67.22	26.12	40.31
PRJNA248621	human islets	22	34,362.72	0.98	9.43	74.55	10.00	31.33
PRJNA540989	pancreatic ductal adenocarcinoma	20	450.69	0.01	5.55	81.70	10.75	49.63
PRJNA34535	NIH Epigenomics Roadmap Initiative	19	51,300.19	4.12	14.74	99.95	0.00	58.24
PRJNA402080	human islet	19	103,070.06	9.11	10.35	62.89	24.00	70.05
PRJNA280220	islet cell	18	27,584.13	0.81	9.22	61.41	31.24	46.96
PRJNA287037	insulin secretion and diabetes risk	18	176,313.01	32.36	11.72	65.41	18.33	53.76
PRJNA553683	Islets	18	12,790.08	0.69	8.09	76.54	17.54	59.05
PRJNA703993	pancreatic islets	18	38,803.35	7.78	5.83	60.14	36.83	57.70
PRJNA497610	pancreatic beta cells	16	126,692.88	2.82	5.89	63.43	26.46	23.07
PRJNA691365	pancreatic β cells	16	316,037.03	57.23	3.31	52.65	33.10	40.37
PRJNA752997	type 1 diabetes	16	209,389.83	19.22	9.85	76.84	10.23	36.60
PRJNA484008	pancreatic neuroendocrine tumors	16	36,626.03	5.90	17.18	64.55	17.13	43.88
PRJNA490335	pancreatic cancer	16	3706.33	0.19	3.75	93.42	6.48	78.33
PRJNA716264	patient-derived organoids	10	1023.80	0.15	19.79	64.31	25.89	64.08
PRJNA630983	pancreatic ductal adenocarcinoma	7	116.67	0.01	55.89	35.33	42.66	53.13

## Data Availability

A compiled file of raw data from figures is available at https://doi.org/10.6084/m9.figshare.21597261.v1 (accessed on 6 December 2022). This file contains the following eight tabs: All Variants VEP: All gnomAD variants around INS with Ensembl Variant Effect Predictor data analysis; INS Regulation Var: The annotation of various regulation datasets for variants around INS; INS Amino Acids: conservation and variation data for insulin amino acids; Pancreas BioProject: compiled analysis of RNAseq projects from pancreas; Pancreas Samples: individual RNAseq samples of pancreas; Liver BioProject: compiled analysis of RNAseq projects from liver; Liver Samples: individual RNAseq samples of liver; INSR Amino Acids: conservation and variation data for insulin receptor amino acids. The gnomAD allele frequencies for variants around INS can be found at https://doi.org/10.6084/m9.figshare.21917124.v1 (accessed on 6 December 2022). Variants are ranked based on the highest frequency for one population group. Columns are colored based on the following key: gray: variant details; yellow: variant annotations; orange: total gnomAD 3.1.2 allele counts from all populations; red: highest allele frequency and population group with that frequency; cyan: allele frequencies for different populations. BLAST analysis of INS and INSR structures within the PDB at https://doi.org/10.6084/m9.figshare.21919194.v1 (accessed on 6 December 2022). Protein alignment files can be found at https://doi.org/10.6084/m9.figshare.21923289.v1 (accessed on 6 December 2022). All figures from the paper are available as original TIFF files at https://doi.org/10.6084/m9.figshare.21923781.v1 (accessed on 6 December 2022).

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
