# Peer review of "Understanding Insulin in the Age of Precision Medicine and Big Data: Under-Explored Nature of Genomics"

_biomolecules, 2023, doi:10.3390/biom13020257_

Round 1
Reviewer 1 Report
The review by Cook et al is a detailed chronicling of the current state genomic data and literature related to insulin. The study does an impressive job of highlighting the key genomic variants associated with disease. The archiving is timely, while incorporating and annotating data for variants from several different databases. I would therefore recommend the paper for publication. The authors should consider working on the following suggestions.
1) At several locations, the authors mention several variants and the population in which the variants are prevalent. The paper, and the general genomic literature around diabetes could benefit siginficantly if the authors could tabulate the population genetic data for the different variants and the subpopulations they are prevalent in.
2) The readability of the article could benefit significantly from a schematic figure that highlights the key steps in insulin synthesis (pre-pro-insulin to pro-insulin to insulin) with the potential error-prone steps at each of these stages.
3) The authors mention potential incorrect disulfide bonds and associated aggregation disorders. Are there clinical/pathological examples of incorrect disulfides. If so, can the authors highlight the most commonly observed erroneous disulfide bridges that are correlated with disease.
Author Response
We thank the reviewer for their support of this paper and their great suggestions to improve. Below are detailed responses to each suggestion.
1) At several locations, the authors mention several variants and the population in which the variants are prevalent. The paper, and the general genomic literature around diabetes could benefit siginficantly if the authors could tabulate the population genetic data for the different variants and the subpopulations they are prevalent in.
Response: We have added a new supplemental table and the following sentence to section 2.1, “The gnomAD allele frequency for different population backgrounds of these variants is available at https://doi.org/10.6084/m9.figshare.21917124.v1.” We have also added a description of this file into the data availability statement.
2) The readability of the article could benefit significantly from a schematic figure that highlights the key steps in insulin synthesis (pre-pro-insulin to pro-insulin to insulin) with the potential error-prone steps at each of these stages.
Response: We attempted to graphicly summarize the steps of insulin synthesis in Figure 11. To reflect this point, we have added the following sentence within section 3, “Nearly every one of the steps within Figure 11 is error-prone and a potential for genetic and environmental interactions to impact Insulin synthesis.”
3) The authors mention potential incorrect disulfide bonds and associated aggregation disorders. Are there clinical/pathological examples of incorrect disulfides. If so, can the authors highlight the most commonly observed erroneous disulfide bridges that are correlated with disease.
Response: We have added into section 2.6 to address loss of disulfide, “Three disulfide bonds maintain Insulin structure [148], where variants of 3/6 cysteines are found pathogenic or likely pathogenic within ClinVar for neonatal diabetes (C43G-VCV000021114.5, C96S- VCV000068730.1, C96Y- VCV000013387.6 , C96R- VCV000918067.1, C109F- VCV001526011.1).” To address incorrect disulfides we added, “One of the primary impairments of Pro-Insulin folding is the formation of intramolecular disulfide alternative pairing, often through the formation of A6/A11 with A7/B7 mispairing, resulting in aggregated fibrils”
Reviewer 2 Report
This comprehensive review focuses on genetic variants of the human insulin gene and their relationship to disease. The authors provide detailed analysis of certain regions of the gene deemed to be as yet fully investigated with respect to genetic variation and take advantage of the ever-increasing data sets generated from whole genome and transcriptome sequencing. The review is well written and provides new insights. The manuscript could be shortened as the Discussion does repeat information provided in the beginning.
Some corrections/suggestions are as follows:
Page 15 line 532 The articles from the Weiss group argue that residue Y50 is important for BOTH receptor binding and also for self assembly. Weiss suggests that there is a convergence between the role of residues in hormone activity and their involvement in stability and folding. He does not argue that involvement in dimer formation is more important than receptor binding as is implied here.
Page 17 line 580 Must include here FB24, FB25, YB26, GA1, IA2, VA3, VB12 in site 1
Page 20 line 644 Please note that while B24-B26 is important for dimer formation, the dimer interface is defined as involving the end of the B chain and particularly residues PB28, KB29. It is a little misleading to state that the dimer region is B24-B26. Actually the residues B23, B24-B26 are involved in formation of a turn that acts as a hinge that allows the end of the B chain to fold away from the core. If the hinge is perturbed, then the end of the B chain folds away from the core and thereby disrupts dimer formation
Figure 1 legend Please acknowledge the genome browser used to generate this figure. There are several panels which use colour for annotation. These are not defined in the legend and should be.
Figure 2 legend Panel A annotation is unclear and very small. What does green shown? In C what is the red box showing? It is unclear what is meant by "cyan box for both SNPs”The font in D is very hard to read
Figure 3 The annotation of regions E, F and G below figure 3C is somewhat confusing. Suggest using a smaller font so these are not confused with the panel annotations
Figure 6, 7 and 8 legends What do the colors in the table show? Pink, red blue??
Figure 8 Panel B would be improved by adding some annotation of the residues
Figure 9 At least mention in the figure legend whether these 232 species include vertebrates and invertebrates, mammalian and non mammalian species? F26Y is conserved across mammalian species and so the conservation score of 1 indicates the analysis must include non mammalian species. Ideally provide a list of species used to derive this information in a supplementary file
Figure 10 (A). The red amino acid sequences are not annotated or described in the legend. Do they correspond to the two ENST sequences at the top? What is meant by "INS loss homology"?
10 (B) It is not clear what panel B is adding. Are you concluding that there is only one match and that is to Pogo abelli INS-IGF2? (the panel D is overlapping panel B here)
Minor corrections/suggestions:
Page 2 line 49 “has been identified as the most significant loci for T1D risk when masking the large effect size HLA loci”. Do you mean second most significant locus after the contribution of the HLA loci? (NB loci is plural, locus is singular)
Page 2 line 68 “modify disease course but do not provide” should read “modify disease course but they do not provide”
Page 2 line 82 “deserts” not “desserts”
Page 3 line 95/96 first sentence does not make sense – “and food deserts live many of the population groups with high risk of diabetes”??
Page 3 line 130 “start sites (TSS) and a known merged transcript” should read “start sites (TSS) and there is a known merged transcript
Page 4 line 155 The phrase “We pulled genomic variants…” is not standard terminology. Do you mean “pooled” or do you mean “extracted” – this needs to be clarified here and later in the Methods section
Page 4 line 175 “started Insulin within one-year diagnosis” would read better as “individuals who started Insulin within one-year diagnosis”
Page 11 line 408 “ENST00000397262 is likely the result of rs689” What is meant by "the result of"? - more detail required
Page 17 line 557 “concentration”
Author Response
We thank the reviewer for their support of this paper and their great suggestions to improve. Of major note, based on the multiple comments for the structural insights from reviewers 2 and 3, we have significantly modified Figures 6-8 and reorganized the text. Our goal was to change the writing to reflect the use of structure and protein interactions in interpreting the missense variants we have from the variant review and not making larger claims on the complexity of the Insulin dynamics. Below are detailed responses to each suggestion.
Page 15 line 532 The articles from the Weiss group argue that residue Y50 is important for BOTH receptor binding and also for self assembly. Weiss suggests that there is a convergence between the role of residues in hormone activity and their involvement in stability and folding. He does not argue that involvement in dimer formation is more important than receptor binding as is implied here.
Response: Changed to, “Alteration of Y50 has revealed that the conservation of the aromatic triplet is linked to dimer (and higher order hexamer) formation and receptor activation.”
Page 17 line 580 Must include here FB24, FB25, YB26, GA1, IA2, VA3, VB12 in site 1
Response: We have updated the text and figure to reflect three different binding events captured in protein structures. The discussion now only addresses the variants and not all amino acids for interaction.
Page 20 line 644 Please note that while B24-B26 is important for dimer formation, the dimer interface is defined as involving the end of the B chain and particularly residues PB28, KB29. It is a little misleading to state that the dimer region is B24-B26. Actually the residues B23, B24-B26 are involved in formation of a turn that acts as a hinge that allows the end of the B chain to fold away from the core. If the hinge is perturbed, then the end of the B chain folds away from the core and thereby disrupts dimer formation
Response: We have updated the contact sites in figure 7, text referencing contacts as observed in the hexamer structure, and text of the model systems.
Figure 1 legend Please acknowledge the genome browser used to generate this figure. There are several panels which use colour for annotation. These are not defined in the legend and should be.
Response: Added, “(USCS, https://genome.ucsc.edu/)”. Regarding color descriptions, this is a tough one as there are so many colors in each panel. Instead of describing each of these we have added the following, “All panel colors from the UCSC genome browser extractions can be found within track information within the browser as we have used default color schemes from the browser.”
Figure 2 legend Panel A annotation is unclear and very small. What does green shown? In C what is the red box showing? It is unclear what is meant by "cyan box for both SNPs”The font in D is very hard to read
Response: For Figure 2A we added, “with green lines representing the gene location.” Figure 2C red box is already labeled in the figure legend as, “The more significant value is shown in red”. To address any potential concerns in the readability of the figures we have uploaded all high-resolution, stand-alone figures and made note of this in the data availability section.
Figure 3 The annotation of regions E, F and G below figure 3C is somewhat confusing. Suggest using a smaller font so these are not confused with the panel annotations
Response: This has been updated within the figure.
Figure 6, 7 and 8 legends What do the colors in the table show? Pink, red blue??
Response: These figures were redone and focus was placed on explaining the coloring code used for all panels.
Figure 8 Panel B would be improved by adding some annotation of the residues
Response: This figure has been redone with primary focus on annotation of variants to receptor binding potential.
Figure 9 At least mention in the figure legend whether these 232 species include vertebrates and invertebrates, mammalian and non mammalian species? F26Y is conserved across mammalian species and so the conservation score of 1 indicates the analysis must include non mammalian species. Ideally provide a list of species used to derive this information in a supplementary file
Response: Changed to, “The conservation is shown as the percent of 232 vertebrate species.” Added into the Data Availability statement, “Alignment files can be found at https://doi.org/10.6084/m9.figshare.21923289.v1.”
Figure 10 (A). The red amino acid sequences are not annotated or described in the legend. Do they correspond to the two ENST sequences at the top? What is meant by "INS loss homology"?
Response: Added, “The nucleotide sequence on top is from ENST00000356578 and the bottom ENST00000397270. INS Loss Homology site is where the INS-IGF2 diverges from INS sequence.”
10 (B) It is not clear what panel B is adding. Are you concluding that there is only one match and that is to Pogo abelli INS-IGF2? (the panel D is overlapping panel B here)
Response: We have fixed the overlap issue. We added the following text to the manuscript, “This uniqueness allows for use of the sequence within human RNAseq extraction in-sights, as no other human transcript matches it with this level of statistical cutoffs.”
Minor corrections/suggestions:
Page 2 line 49 “has been identified as the most significant loci for T1D risk when masking the large effect size HLA loci”. Do you mean second most significant locus after the contribution of the HLA loci? (NB loci is plural, locus is singular)
Response: We have changed this to, “In multiple independent studies, the rs689 variant of the insulin gene (INS) has been identified as the most significant locus for T1D risk when masking the large effect size HLA loci.” Contrary to the Insulin literature, genetics rarely reports HLA loci now for GWAS due to the complications in genotyping and need to refine based on newer T2T genome. With this, the locus is often masked now in data reporting for SNP chips as in the case of most data access to T1D GWAS. Thus, this SNP is the most significant from data view point. As proof of this point, I refer the reviewer to https://genetics.opentargets.org/study/GCST90014023 and https://genetics.opentargets.org/study/GCST010681
Page 2 line 68 “modify disease course but do not provide” should read “modify disease course but they do not provide”
Response: changed as suggested
Page 2 line 82 “deserts” not “desserts”
Response: changed as suggested
Page 3 line 95/96 first sentence does not make sense – “and food deserts live many of the population groups with high risk of diabetes”??
Response: Changed to, “Many of the population groups with high risk of diabetes Within many of theselive within communities disproportionately impacted by urban planning, air pollution, low income, and food deserts live many of the population groups with high risk of diabetes.”
Page 3 line 130 “start sites (TSS) and a known merged transcript” should read “start sites (TSS) and there is a known merged transcript
Response: changed as suggested
Page 4 line 155 The phrase “We pulled genomic variants…” is not standard terminology. Do you mean “pooled” or do you mean “extracted” – this needs to be clarified here and later in the Methods section
Response: changed as suggested
Page 4 line 175 “started Insulin within one-year diagnosis” would read better as “individuals who started Insulin within one-year diagnosis”
Response: we prefer this left as is with the term identical to the data reporting of UKBiobank, which allows for better searching if others want to follow up on this. See https://genetics.opentargets.org/study/NEALE2_2986
Page 11 line 408 “ENST00000397262 is likely the result of rs689” What is meant by "the result of"? - more detail required
Response: Changed to, “This suggests that due to the high penetrance of signal in RNAseq, the alteration of the 5’UTR splicing change of ENST00000397262 is likely the result of rs689.”
Page 17 line 557 “concentration”
Response: changed to level
Reviewer 3 Report
The manuscript review by Cook et al. titled “Understanding Insulin in the Age of Precision Medicine and Big Data: Under-Explored Nature of Genomics.” examines the literature for the genetics which are involved in insulin transcription, processing, translation, protein structure, receptor binding and metabolism. The authors present a relatively wholistic review in this sense, and demonstrate quite a detailed review of the insulin genetics and mutations which occur within the population.
The manuscript review in its current form is relatively well written, and likely of benefit to those concerned with insulin genetics and the factors which influence possible mutations within the populace. In detailing insulin action from the INS gene to receptor binding and subsequent metabolism, the review also describes insulin structure, and more briefly the insulin-insulin receptor (IR) interaction. Although I have identified a number of concerns which the authors should address prior to this work being accepted for publication, I would first suggest the review be either expanded significantly to build on these areas, or, more appropriately the review focus exclusively on insulin genetics (discussed below).
I have provided both major and minor points below to the authors that I believe would strengthen the manuscript review to increase the readability of the work.
Major Concerns
- The literary review of insulin structure and insulin-IR interaction contained here both doe not seem to fit with the principally genetic review, and is limited in scope. This has the effect that it is superficial and provides very limited information to the reader. As an example, there exists a wealth of information describing insulin structure during storage (R-state) and towards activation (T-state) which would have impact on future therapeutics (basal vs fast acting). There is also significant work describing mutations which occur in insulin (eg. Insulin Wakayama, Los Angeles and Chicago) which have structural effects. Where insulin mutations are discussed from the genetic viewpoint in this work, inclusion of insulin structure as written would therefore necessitate a description of how these mutations alter this structure. This is absent.
There is also very little description of the insulin-IR interaction. Noting the extensive and rapidly evolving situation of insulin-IR cryoEM structures within the past 5 years providing a wealth of information, there is very significant scope to expand here should it be relevant. Importantly, the wealth of information from these recent studies describing different stoichiometries (from partially unsaturated to fully saturated receptors) and therefore different receptor orientations is important in understanding the role by which insulin engages. This therein has follow-on effects in how such engagement changes down-stream signalling.
Furthermore, the insulin structure and receptor engagement is only briefly mentioned with the Discussion, and its impact on future personalised medicine is not really addressed. I would therefore suggest that the sections relating to insulin structure and receptor engagement are removed and the work focus exclusively on the INS and associated genes. This will both improve readability and not distract the reader with a superficial and cursory exploration of insulin structure and receptor engagement.
- The figures throughout the work are difficult to read and in some instances unreadable. As an example, Figures 3B, 3D, 3I, and 4A which even at greater than 100% zoom are difficult to interpret. In printed form these would be entirely un-interpretable. I would suggest the authors split these figures where appropriate to improve readability. Also, the grey and black backgrounds of Figure 6 and 7, respectively, are inconsistent with the other Figures.
Minor Concerns
- The peptide insulin is capitalised throughout. This is not necessary.
- There is a grammatical error on line 474, “could be associated diabetes”, should read “could be associated with diabetes”
- In Figure 6 it would be worth noting that this structure is of insulin is in the R-state, and not the typical active T-state. This is important due to the structural re-arangement which occurs to facilitate insulin-IR engagement.
- The term C-peptide vs. C chain is used interchangeably throughout the work - ie. from line 495-501, and then on line 492 and 628. I suggest that it is referred to as the C-chain when discussed as a single-chain peptide, and the C-peptide when cleaved. Also, the B-chain is interchangeably referred to as both “B-chain” and “B chain”. I would suggest uniformity to increase readability.
- Line 524 describes pro-insulin forming a homodimer structure, however it refers to Figure 7 which displays insulin (as stated within the figure title).
- Line 527 continues using the final insulin numbering, however line 528 reverts back to the initial numbering. It would be more succinct to maintain a single convention throughout this work where able - this reviewer would suggest after initial description of full-peptide numbering that final numbering is used to maintain comparison to other work.
- In Figure 7 the Zn atom centre is not described or annotated in the figure or legend.
- In Line 577 and line 579-582, what is meant by the term “allosteric alteration” when referring to IR activation? Also, the residues indicated as essential in engaging IR site 1 or site 2 are incomplete. As an example the aromatic triplet residues (B24-B25-B26) as detailed earlier in the work play an integral role. There are numerous publications describing these interactions, notably the many recent cryo-EM structures, however this was first described in detail in the following, Menting et al., PNAS, 2014.
- Figure 8: Insulin is shown bound to the IR with two peptides bound. It would be worth describing the stoichiometry of the engagement, and whether these relate to which sites (site 1 or site 2) described in the text to help the reader. Although the authors have attempted to make understanding the structural engagement easier to interpret (by having the receptor predominately grey) the colours used here and lack of labelling within the figure make it hard to decipher.
Author Response
We thank the reviewer for their support of this paper and their great suggestions to improve. Of major note, based on the multiple comments for the structural insights from reviewers 2 and 3, we have significantly modified Figures 6-8 and reorganized the text. Our goal was to change the writing to reflect the use of structure and protein interactions in interpreting the missense variants we have from the variant review and not making larger claims on the complexity of the Insulin dynamics. Below are detailed responses to each suggestion.
I would first suggest the review be either expanded significantly to build on these areas, or, more appropriately the review focus exclusively on insulin genetics (discussed below).
Response: This is a great suggestion. Thus we have reformatted figures 6-8 and redone the text to keep more focus on how variants may influence known structures.
- The literary review of insulin structure and insulin-IR interaction contained here both doe not seem to fit with the principally genetic review, and is limited in scope. This has the effect that it is superficial and provides very limited information to the reader. As an example, there exists a wealth of information describing insulin structure during storage (R-state) and towards activation (T-state) which would have impact on future therapeutics (basal vs fast acting). There is also significant work describing mutations which occur in insulin (eg. Insulin Wakayama, Los Angeles and Chicago) which have structural effects. Where insulin mutations are discussed from the genetic viewpoint in this work, inclusion of insulin structure as written would therefore necessitate a description of how these mutations alter this structure. This is absent.
Response: We appreciate this point. Thus, in reworking the manuscript we focused on sequence analysis of the PDB followed by taking several of those structures and interpreting the location of variant amino acids at contact sites. In the discussion, we added the following sentence to better capture our limitations, “While this article focused on the location of current known variants within solved protein structures, there is a wealth of knowledge of the biophysics of Insulin dynamics that needs additional integration into the larger variant potential for variant interpretations.”
There is also very little description of the insulin-IR interaction. Noting the extensive and rapidly evolving situation of insulin-IR cryoEM structures within the past 5 years providing a wealth of information, there is very significant scope to expand here should it be relevant. Importantly, the wealth of information from these recent studies describing different stoichiometries (from partially unsaturated to fully saturated receptors) and therefore different receptor orientations is important in understanding the role by which insulin engages. This therein has follow-on effects in how such engagement changes down-stream signalling.
Response: We have added the receptor PDB analysis with overlap of Insulin to reflect the current structural knowledge insights. In addition, we have added the cryoEM structure of INSR and INS and highlighted variant locations.
Furthermore, the insulin structure and receptor engagement is only briefly mentioned with the Discussion, and its impact on future personalised medicine is not really addressed. I would therefore suggest that the sections relating to insulin structure and receptor engagement are removed and the work focus exclusively on the INS and associated genes. This will both improve readability and not distract the reader with a superficial and cursory exploration of insulin structure and receptor engagement.
Response: The complete removal of this would be detrimental to readers not familiar with Insulin as they would not realize the potential of variants on receptor interaction. We hope our reworking of the figures and text helped to focus the story. Additionally, we added into the discussion the following, “The receptor interaction and metabolism pathways represent one of the most exciting aspects of Insulin precision medicine as it may hold additional answers for therapeutic derivative choice within each individual.”
- The figures throughout the work are difficult to read and in some instances unreadable. As an example, Figures 3B, 3D, 3I, and 4A which even at greater than 100% zoom are difficult to interpret. In printed form these would be entirely un-interpretable. I would suggest the authors split these figures where appropriate to improve readability. Also, the grey and black backgrounds of Figure 6 and 7, respectively, are inconsistent with the other Figures.
Response: We have provided the figures as source files with the ability to view in higher resolution, as added in the data availability section. As with any paper, the viewability of figures is more hinged on how the journal posts, how it is downloaded/opened, and how it is printed. As these are outside of the authors control, source file availability has been the solution for most papers where there is concern. Figure 6 and 7 were updated to look like the others.
Minor Concerns
- The peptide insulin is capitalised throughout. This is not necessary.
Response: According to international standards, all human proteins including those processed should be capitalized. This is reflected in databases such as UniProt. If referring to general insulin it should be lowercase, but here we are referring to the human protein and thus it should be capitalized.
- There is a grammatical error on line 474, “could be associated diabetes”, should read “could be associated with diabetes”
Response: changed as suggested
- In Figure 6 it would be worth noting that this structure is of insulin is in the R-state, and not the typical active T-state. This is important due to the structural re-arangement which occurs to facilitate insulin-IR engagement.
Response: We have updated the text and figure to reflect on NMR structure determination of Pro-Insulin for figure 6 and figure 8 the conformations of Insulin on receptor binding.
- The term C-peptide vs. C chain is used interchangeably throughout the work - ie. from line 495-501, and then on line 492 and 628. I suggest that it is referred to as the C-chain when discussed as a single-chain peptide, and the C-peptide when cleaved. Also, the B-chain is interchangeably referred to as both “B-chain” and “B chain”. I would suggest uniformity to increase readability.
Response: Corrected as suggested.
- Line 524 describes pro-insulin forming a homodimer structure, however it refers to Figure 7 which displays insulin (as stated within the figure title).
Response: Corrected.
- Line 527 continues using the final insulin numbering, however line 528 reverts back to the initial numbering. It would be more succinct to maintain a single convention throughout this work where able - this reviewer would suggest after initial description of full-peptide numbering that final numbering is used to maintain comparison to other work.
Response: Corrected.
- In Figure 7 the Zn atom centre is not described or annotated in the figure or legend.
Response: Updated Figure 7 to correct.
- In Line 577 and line 579-582, what is meant by the term “allosteric alteration” when referring to IR activation? Also, the residues indicated as essential in engaging IR site 1 or site 2 are incomplete. As an example the aromatic triplet residues (B24-B25-B26) as detailed earlier in the work play an integral role. There are numerous publications describing these interactions, notably the many recent cryo-EM structures, however this was first described in detail in the following, Menting et al., PNAS, 2014.
Response: Updated figure and text to address this concern, bringing in the cryo-EM structure.
- Figure 8: Insulin is shown bound to the IR with two peptides bound. It would be worth describing the stoichiometry of the engagement, and whether these relate to which sites (site 1 or site 2) described in the text to help the reader. Although the authors have attempted to make understanding the structural engagement easier to interpret (by having the receptor predominately grey) the colours used here and lack of labelling within the figure make it hard to decipher.
Response: This was updated and a focus within the new Figure 8.
Round 2
Reviewer 3 Report
After reading the revised manuscript by Cook et al., the authors have adequately addressed the majority of concerns I raised, and the manuscript has been enhanced due to the changes made in response to all reviewers.
However, despite providing a response as to why the majority of figures have very small text (less than 4pt in many cases) I very strongly suggest the authors review and make improvement in this work and all work subsequently. Such small text has a negative impact on the readability of the work despite how it is presented, and is the reason why all journals have text size limits for the bulk of the manuscript. The minimum text size required by most/all journals is despite the authors providing the 'source' text file (as a MS word or LaTEX file) of the manuscript and not dependant on 'how the journal posts, how it is downloaded/opened, and how it is printed'.
The remainder of the work is to a good standard and confirm a positive evaluation.